# High-throughput analysis of single human cells reveals the complex nature of DNA replication timing control

Dashiell J. Massey ⬤ [1] & Amnon Koren ⬤ [1✉]

DNA replication initiates from replication origins firing throughout S phase. Debate remains about whether origins are a fixed set of loci, or a loose agglomeration of potential sites used stochastically in individual cells, and about how consistent their firing time is. We develop an approach to profile DNA replication from whole-genome sequencing of thousands of single cells, which includes in silico flow cytometry, a method for discriminating replicating and non-replicating cells. Using two microfluidic platforms, we analyze up to 2437 replicating cells from a single sample. The resolution and scale of the data allow focused analysis of replication initiation sites, demonstrating that most occur in confined genomic regions. While initiation order is remarkably similar across cells, we unexpectedly identify several subtypes of initiation regions in late-replicating regions. Taken together, high throughput, high resolution sequencing of individual cells reveals previously underappreciated variability in replication initiation and progression.

[1] Department of Molecular Biology and Genetics, Cornell University, Ithaca, NY 14853, USA. ✉email: koren@cornell.edu

Faithful duplication of the genome is a critical prerequisite to successful cell division. Eukaryotic DNA replication initiates at replication origin loci, which are licensed in the $G_1$ phase of the cell cycle and fired at different times during the S phase. In many eukaryotes, sequencing of cells at different stages of the cell cycle has been used to profile DNA replication timing, which measures the relative time that different genomic regions are replicated during S phase (reviewed in[1]). This replication-timing program is highly reproducible across experiments[2], suggesting strict regulatory control; and conserved across phylogeny[3,4], suggesting selection under evolutionary constraint. However, the molecular mechanisms that determine the locations and preferred activation times of replication origins in mammalian genomes remain unclear. Furthermore, debate persists over whether the reproducible nature of the replication-timing program reflects the consistent activity across cells of specific individual replication origins or stochastic firing of different origins in different cells within a given region. Ensemble replication-timing measurements have been interpreted to indicate that replication is organized in broad domains, spanning hundreds of kilobases to several megabases, with consistent replication timing governed by the activity of clusters of replication origins[5,6]. Furthermore, some recent replication origin-mapping methods have indicated that replication origins are highly abundant and highly dispersed throughout the human genome[1,7], suggesting that many sites may function as origins used in a subset of cell cycles. In contrast, high-resolution measurements of hundreds of human replication-timing profiles[8,9], or replication timing across multiple S-phase fractions[10], support initiation of replication from more localized genomic regions. While these replication-timing methods reveal genomic regions that reproducibly replicate at characteristic times during S phase, it remains contested whether these represent a conserved pattern across cells or reflect the average behavior of single cells. Previous work has modeled how the stochastic firing of replication origins could be sufficient to explain the replication-timing profile[11,12], and single-molecule experiments (e.g., with DNA combing) have suggested that cells may use different subsets of origins in each cell cycle[13,14].

Recently, replication timing has been analyzed by single-cell sequencing of several hundred mouse or human cells[15–17]. These studies focused on cells in the middle of S phase and analyzed replication at the level of domains, concluding that stochastic variation exists in replication timing and is the highest in the middle of S phase. However, single-molecule and single-cell studies have been limited in their throughput and biased toward early S-phase or mid-S-phase, respectively. Analyzing many cells is particularly important, given that even when the whole genome is captured, a single cell provides only a snapshot of DNA replication at a single moment in time. By assaying many cells at different stages of S phase, it is possible to string these snapshots together to construct a picture of replication states over time. However, the resolution of this picture will be dependent both on capturing cells at many stages of S phase and on assaying a large number of cells.

Here, we report the analysis of whole-genome sequencing of thousands of single replicating cells across ten human cell lines. We developed an in silico approach to sort cells by cell cycle state, allowing us to capture cells throughout the full duration of S phase, and to analyze them in any number of sub-S-phase fractions down to single-cell resolution. We found that single cells within a given cell line largely used a consistent set of replication-initiation regions, which were discrete genomic loci rather than megabase-scale domains. Furthermore, these initiation regions fired in a predictable, albeit not fixed, order. Some initiation regions were consistently fired early in S phase across cells, while others were fired consistently late. However, we also identified a subset of rarely fired initiation regions with a preference for early firing and another subset that fired throughout S phase. We conclude that a consistent set of replication origins explains the vast majority of replication-initiation events in single cells, and that existing models of replication timing fall short of explaining the diversity of firing-time patterns.

## Results

**High-throughput measurement of single-cell replication.** Previous sequencing-based studies measured DNA replication timing in a relatively small number of cells, mostly limited to mid-S-phase cells[15,16]. To analyze single cells, these studies performed DNA amplification using degenerate oligonucleotide-primed PCR (DOP-PCR), with one reaction per cell. Due to technical noise introduced during amplification, these studies were limited to analyzing replication timing at the level of large chromosomal domains (typically on the order of megabases). As an alternative approach, we devised a method to study DNA replication timing across the entire span of S phase, in hundreds to thousands of cells, and with higher spatial resolution than previous methods. Specifically, we used two microfluidic systems that isolate and barcode single-cell DNA: the 10x Genomics Single Cell CNV platform, which performs multiple-displacement amplification (MDA) on pooled barcoded cells, and direct DNA transposition single-cell library preparation (DLP+)[18], which is an amplification-free method. Both library-preparation methods were followed by whole-genome sequencing of single cells. With each platform, automation of labor-intensive steps allows for dramatic increases in throughput. In addition, recent studies suggest that improved MDA protocols may have reduced noise relative to previous single-cell amplification methods[19].

As an initial proof-of-principle, we analyzed 5793 cells from the human lymphoblastoid cell line (LCL) GM12878 isolated with the 10x Genomics system, following fluorescence-activated cell sorting (FACS) of $G_1$-, $G_2$-, and several fractions of S-phase cells. The resulting sequencing data were sufficient to distinguish replicating cells from nonreplicating cells across a fivefold range of sequencing read depths (50–250 reads per Mb). Specifically, local read depth fluctuated more in replicating cells relative to nonreplicating cells of similar coverage (Fig. 1a). To validate that these fluctuations could be used to computationally distinguish replicating cells from nonreplicating cells within an unsorted population, we quantified them using MAPD (median absolute deviation of pairwise differences between adjacent genomic windows[20]), which scales proportionally to read depth (Methods). Indeed, FACS-sorted $G_1$- and S-phase cells had distinct linear relationships between scaled MAPD and average read depth (Fig. 1b). Therefore, we were able to computationally assign each cell as "$G_1$" or "S" (Fig. 1b), and compare the resulting fractions to the FACS labels. In silico sorting was highly concordant with FACS labels (Fig. 1b; Supplementary Fig. 1), allowing us to perform additional experiments without FACS. Accordingly, we sequenced an additional three GM12878 samples without cell sorting, recovering an additional 3787 cells in total. We analyzed these cells together with the sorted cell libraries, as described below.

Post hoc in silico cell sorting from single-cell sequencing has two major benefits over sequencing single cells isolated from multiple flow cytometry-sorted populations. First, sequencing biases (particularly, GC-content bias[21]) are known to vary between sequencing libraries, a concern alleviated by using control cells from within the same library as the cells of interest. Second, this approach minimizes experimental manipulations, does not require DNA staining, and reduces interexperimental variation, for instance, in defining FACS gates. However, other

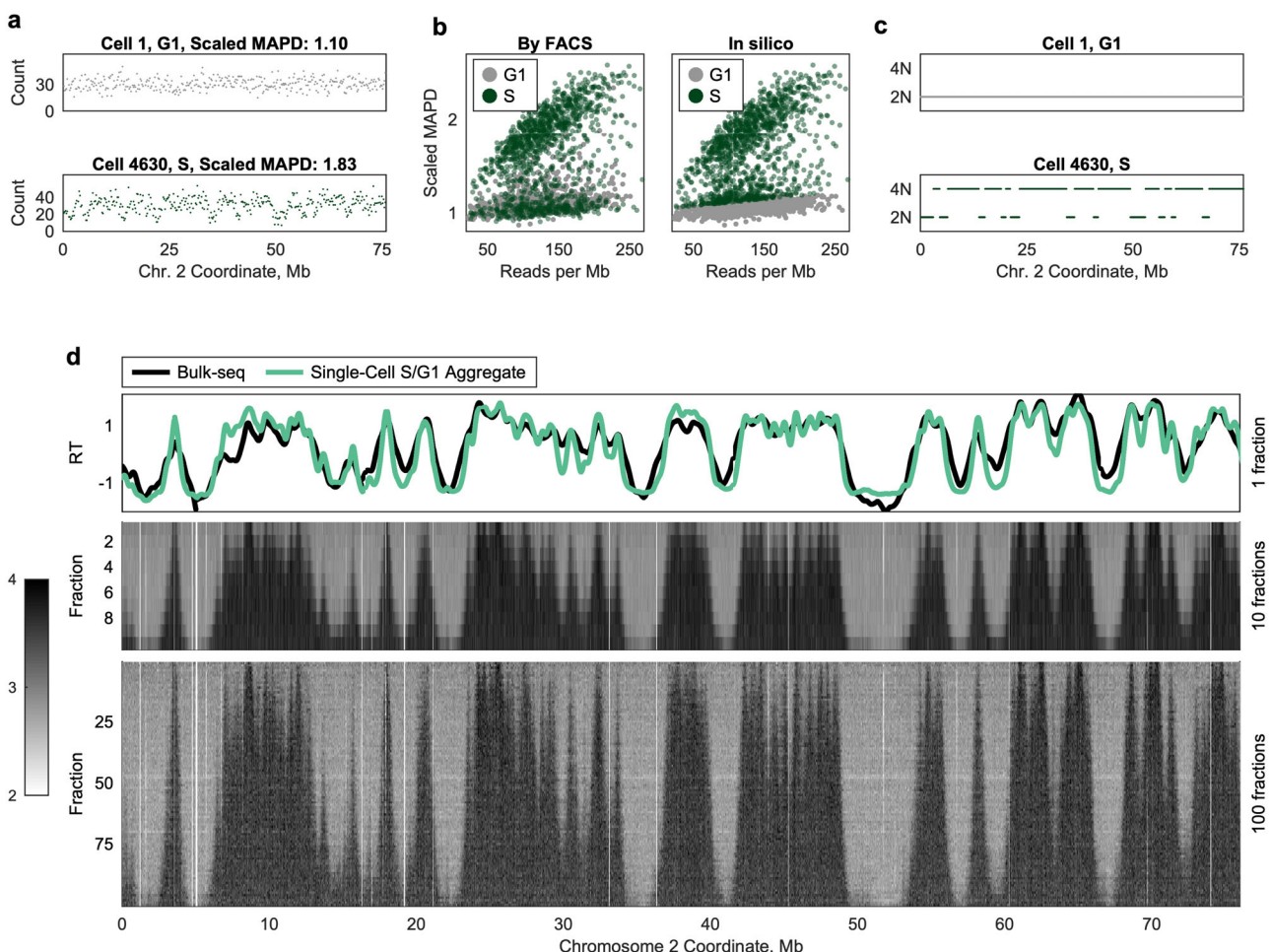

**Fig. 1 Discrimination of replicating and nonreplicating cells by in silico flow cytometry. a** Nonreplicating $G_1$-phase cells (e.g., Cell 1) display relatively uniform read depth across the genome, whereas S-phase cells (e.g., Cell 4630) display fluctuations in read depth, consistent with two underlying copy-number states. Dots represent read count in 200 kb windows. **b** Flow-sorted single cells (*left*) can be accurately sorted in silico (*right*). Replicating S-phase cells display a higher degree of read-depth fluctuation relative to nonreplicating $G_1$-phase cells sequenced to equivalent coverage (quantified by scaled MAPD, median absolute pairwise difference between adjacent genomic windows). *Left*: cells are labeled as $G_1$- (gray) or S-phase (green) based on FACS. Only $G_1$- and S-phase fractions are shown. *Right*: the same cells are labeled as $G_1$ or S based on scaled MAPD, revealing widespread cross-contamination. **c** Replication profiles were inferred for each single cell, using a two-state hidden Markov model. Nonreplicating cells (e.g., Cell 1) display a single copy number (2N), while replicating cells (e.g., Cell 4630) display two distinct copy-number states (2N, 4N). Dots represent inferred replication state in 20 kb windows. The same region is shown from **a**. **d** Aggregating data across S-phase cells into one or more fractions reveals a consistent structure of replication progression at different times in S phase. *Top*: ensemble replication timing inferred from all S-phase cells (green) was highly correlated with bulk-sequencing replication timing for the same cell line (black). *Middle, bottom*: single cells were aggregated into 10 or 100 fractions based on S-phase progression. Triangular pileups of high read depth (caused by replication in most/all cells in the fraction) are observed in discrete locations across the chromosome, suggesting bidirectional replication progression originating at fixed loci. Each row represents one fraction (containing multiple cells). Columns represent 20 kb windows. Low-mappability regions (white) have been masked.

strategies may be more cost-effective: typical mammalian cell cultures contain up to ~30% of cells in S phase at any given time.

Using a conservatively defined subset of nonreplicating cells identified by this in silico cell-sorting approach, we defined variable-size, uniform-coverage genomic windows that accounted for the effects of mappability and GC-content biases, as well as copy-number variations, on sequencing read depth[22]. (We note that in silico sorting cannot distinguish $G_2$ cells from $G_1$ cells because, in principle, both have a uniform copy number genome-wide. We will therefore refer to these cells as "$G_1/G_2$ cells" throughout.) We counted the number of sequencing reads in each window for each cell (Supplementary Fig. 2), and then used a two-state hidden Markov model (HMM) to infer whether each window contained replicated or unreplicated DNA (Methods). This confirmed the uniform DNA copy number across the

genome in $G_1/G_2$ cells, and fluctuating regions of replicated and unreplicated DNA in S-phase cells (Fig. 1c). We further validated the HMM by simulation, estimating that, on average, 96.4% of 20-kb windows were called accurately in each cell (Supplementary Fig. 3).

Our proof-of-principle FACS experiment also revealed cross-contamination between fractions: in silico sorting labeled ~24.3% of cells in the $G_1$-phase FACS fraction as "S" and reciprocally ~25.1% of cells in the S-phase fractions as "$G_1/G_2$" (Supplementary Fig. 1). However, because the objective of in silico sorting is to identify high-confidence $G_1/G_2$ cells to use as controls, it is designed to be conservative in labeling cells as "$G_1/G_2$". We therefore suspected that this estimate of S-phase cells in the $G_1$ FACS fraction was inflated. Indeed, after HMM processing, 260/323 cells in the $G_1$ fraction initially called as "S" were reassigned

as G$_1$/G$_2$. The remaining 63 cells (4.7% of the G$_1$ fraction) were confirmed S-phase cells, displaying copy-number profiles consistent with early S-phase and indistinguishable from early S-phase cells in the S-phase fraction (Supplementary Fig. 1). We further analyzed a published dataset of 1475 cells from the human melanoma cell line COLO-829, for which FACS was used to isolate exclusively G$_1$ cells[23]. In this dataset, in silico sorting labeled 233 cells (15.8%) as "S", of which 32 cells were confirmed to be in S-phase (cross-contamination: 2.2%; Supplementary Fig. 4). Thus, in silico sorting using single-cell sequence DNA is a viable strategy for identifying control (G$_1$/G$_2$) cells within an unsorted library, and when used in combination with the HMM, provides greater sensitivity to FACS.

A discrete benefit of single-cell data is the ability to aggregate similar cells together, effectively increasing the coverage without masking important heterogeneity between subsets of cells. Because the partitioning happens in silico, we can consider many different single-cell aggregates of the same data, from a single fraction (spanning all of S phase) down to single cells (wherein each cell is its own fraction). We generated several such aggregates, partitioning cells based on their progression through S phase (% of genome replicated) and summing per-window read counts across cells (Fig. 1d). Validating this approach, the single-fraction profile—analogous to an ensemble S/G$_1$ replication-timing profile[22]— was highly correlated to a bulk replication-timing profile for the same cell line ($r = 0.90$). Partitioning cells into 10 fractions, a structure emerged similar to that seen in high-resolution Repli-seq[10]: triangular pileups of high read depth (corresponding to active replication) around peaks observed in bulk sequencing. Many of these regions of high read depth were evident in every fraction, although some (e.g., Fig. 1d, ~13.8 Mb) first appear later in S phase. This same structure was observed—at higher resolution—when cells were partitioned into 100 fractions. Thus, by this approach, we can capture sub-S-phase events across all of S phase without the risk of FACS cross-contamination, at a resolution for which FACS is infeasible (i.e., 100 fractions), and with the ability to examine the same population of cells at multiple levels of resolution.

The logical extension of this partitioning approach is to consider each cell as comprising its own fraction. After filtering out cells that were not replicating or for which a twofold relationship was not observed between copy-number states, we analyzed 2437 single GM12878 cells. At this single-cell resolution, we observed consistent pileups of distinct replicated and unreplicated segments across cells (Fig. 2a). These pileups were in the same regions observed as peaks in the bulk-sequencing profile and sub-S-phase fractions, underscoring that these regions correspond to locations of active replication progression, centered at one or more replication origins. Even at single-cell resolution, these pileups were triangular (Fig. 2a, insets), consistent with symmetric bidirectional replication-fork progression from a common origin locus (or a tight cluster of replication origins), and appeared visually to be highly localized. Thus, we demonstrate the ability to measure single-cell replication timing in thousands of single cells, in an unbiased manner, and without the need for FACS. This represents roughly ten times more cells than have been reported in previous single-cell replication-timing analyses, which have focused primarily on mid-S-phase cells[15,16].

We repeated this analysis using single-cell data for 3040 cells from the LCL GM18507, prepared using DLP+[18]. We identified 759 replicating cells within this dataset, and again observed pileups in consistent genomic regions, close to peaks in the S/G$_1$ aggregate replication-timing profile (Fig. 2b). This dataset enabled us to benchmark our analysis strategy in the absence of amplification bias, ensuring that the observed single-cell pileups were not a persistent technical artifact of the 10x Genomics amplification method and validating the ability to accurately profile single-cell replication timing in hundreds to thousands of cells across multiple single-cell sequencing technologies.

**Sites of replication initiation are consistent in single cells.** The nature of DNA replication initiation events is among the most debated aspects of mammalian DNA replication, both regarding its spatial scale (specific loci[24–27], localized regions[28–30], or broad domains[5,6]) and the degree of spatial and temporal stochasticity across cells[1,11,12]. Our comprehensive single-cell DNA replication data enable us to rigorously address these subjects.

We focused first on the spatial dimension of variability among cells. As noted above, visual inspection of replicated-region pileups revealed very little variation across single cells (Fig. 2; Fig. 3a). To analyze this axis of variation systematically, we began by identifying replicated segments in each single cell. Each replicated segment, which we termed a track (by analogy to single-molecule DNA combing tracks), represents the activity of at least one replication origin. Theoretically, if a replication track corresponds to a single replicon, initiating from one origin and expanded by symmetric progression of sister replication forks, the origin of replication should be located at the center of that replication track. Thus, as a first approximation of origin locations, we assigned the midpoint of each replication track as the most likely location of replication initiation for that track. (We excluded tracks longer than 1 Mb in this initial analysis to reduce the likelihood of including tracks that reflected the activity of multiple independent origins that have converged.)

Consistent with previous work suggesting that replication-initiation potential is diffuse throughout the genome[7], we found that 49.7% of mappable 20kb genomic windows were called as a probable initiation site in at least one cell. However, these probable initiation sites were not uniformly distributed across the genome. Rather, highly frequent initiation sites were neighbored by gradually less frequent initiation sites, creating peaks around these local maxima (Fig. 3a). This structure suggests that a more limited group of genomic loci might give rise to replication initiation, as ambiguity in identifying the boundaries of replication tracks would result in slight shifts in the probable initiation site from the true midpoint to a neighboring locus and the observed gradual decrease in initiation frequency with increasing distance from that true midpoint.

Based on the conclusion that noise in individual cells was likely contributing substantially to variation in initiation-site location, we devised an approach to cluster these sites into larger initiation regions (IRs) shared across cells, which did not rely on a 1-Mb length cutoff to determine which replication tracks were informative about individual origins and which represented the activity of multiple independent origins. Instead, replication tracks that overlapped multiple shorter replication tracks were treated as agnostic to IR location because they could plausibly be explained by firing of multiple of the overlapped origins or of a single central origin. These uninformative tracks were thus excluded from use in clustering. By this process, we identified a total of 7522 IRs.

As noted above, single-cell pileups corresponded visually to peaks in the S/G$_1$ aggregate replication-timing profile (Fig. 2; Fig. 3a). Indeed, 90.9% of peaks in the aggregate profile coincided with an IR. Of these aggregate peaks that overlapped an IR, 48.7% corresponded to multiple IRs (e.g., Fig. 3b, left), while the remaining 51.3% corresponded to a single IR (e.g., Fig. 3b, right). This suggests that origins are often clustered in hotspots along the chromosome; the replication-timing peaks corresponding to single IRs could either be regions of lower origin density or, conversely, represent hotspots too dense for individual origins to

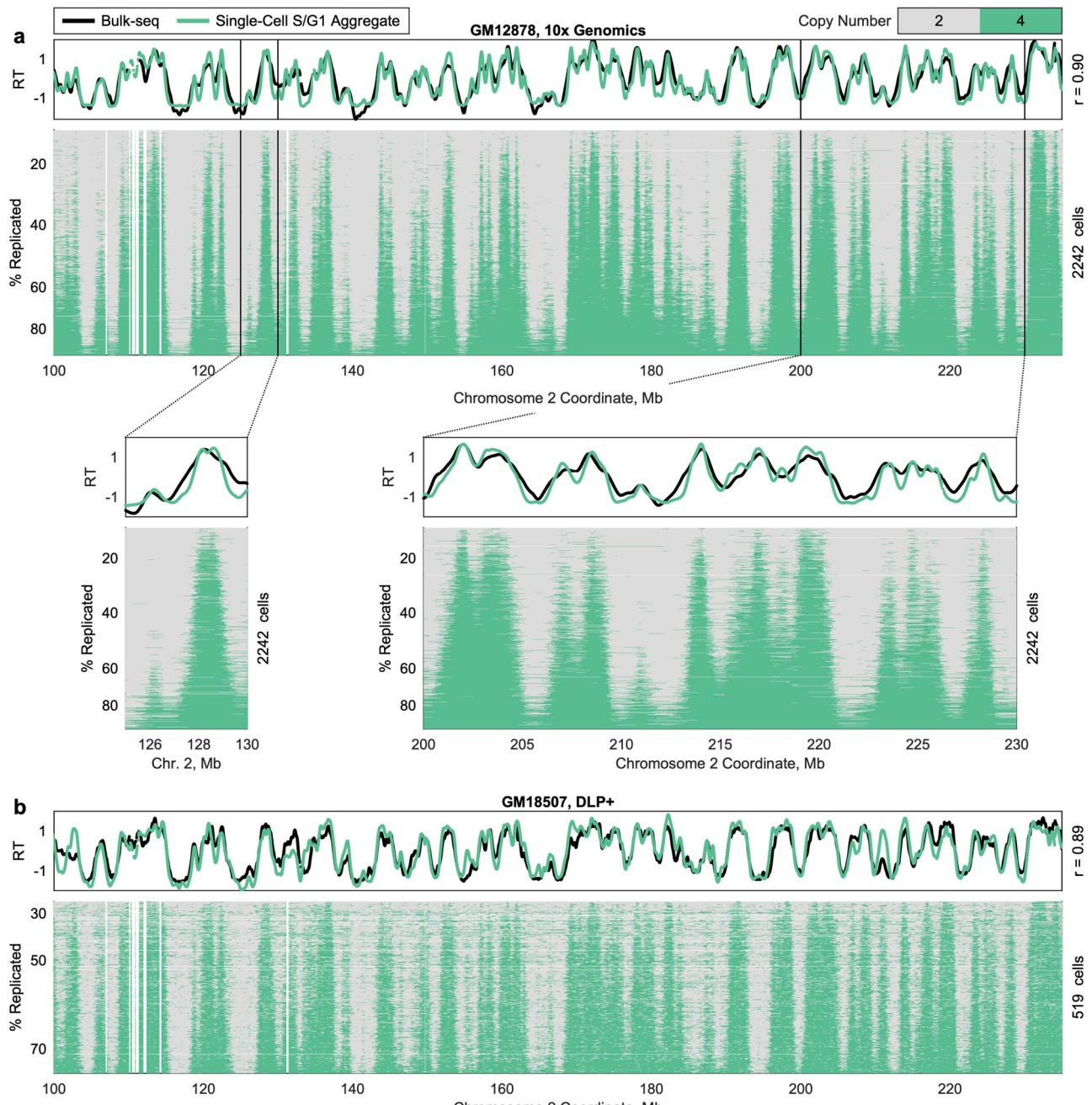

**Fig. 2 Single-cell replication-state data, generated by multiple library preparation protocols. a** Single-cell replication profiles for 2242 GM12878 cells (including both sorted and unsorted cells), following single-cell isolation and library preparation with the 10x Genomics Single-Cell CNV Solution. Consistency of the replication program is observed across cells at chromosome scale and at the level of individual peaks (inset). Pileups reflect sharply defined and consistently replicated regions, which overlap peaks in the bulk replication-timing profile. Variation in activation time during S phase among initiation sites is also observed to mirror the replication-timing profile. Each row represents a single cell, sorted by the percent of the genome replicated, and each column represents a fixed-size window of 20 kb. In total, 195 cells are not shown due to copy-number aberrations on this chromosome. Low-mappability regions and cell-specific copy-number alterations have been masked (white). Insets show smaller regions. **b** Single-cell replication profiles for 519 GM18507 cells, following amplification-free direct DNA transposition single-cell library preparation (DLP+). Due to noise, only 480–614 of the 759 S-phase cells were analyzed for any given chromosome. Raw data are from[18].

be detected at this resolution. Thus, single-cell data are concordant with the ensemble replication-timing profile, but also caution that smoothing of ensemble profiles likely removes information about distinct initiation sites.

We then assigned all replication tracks shorter than 1 Mb to the IR whose center was closest to the midpoint of the track. This includes tracks that potentially overlap multiple fired IRs;

however, when all replication tracks assigned to a given IR were sorted by length, a symmetric triangle was observed around the IR center (Fig. 3c), consistent with sister replication forks progressing away from a single origin or tight cluster of origins at the IR center with similar processivity. For each IR, we calculated how tightly the midpoints of these replication tracks were clustered to assess how precisely the most probable

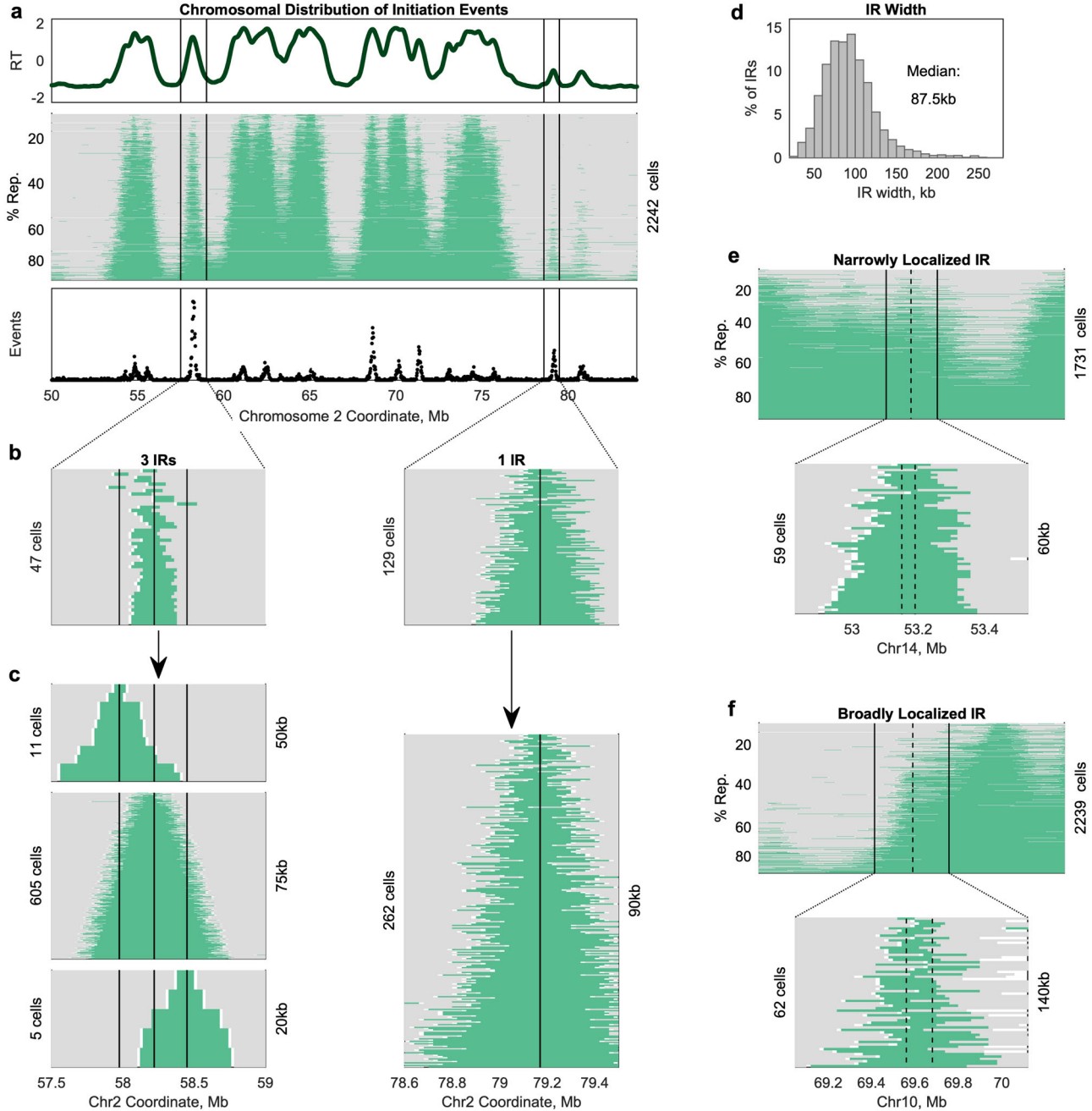

**Fig. 3 Consistency of single-cell replication-initiation sites. a** Peaks in the aggregate replication-timing profile inferred from all GM12878 S-phase cells (*top*) correspond to consistently replicated segments across single cells (*middle*), and to regions of dense initiation-site calls (*bottom*). **b** Replicated regions in single cells are centered at consistent locations, termed initiation regions (IRs), overlapping peaks in the aggregate replication-timing profile. For each IR (black line), a subset of cells contained a replicated track (green) overlapping the IR center but not extending into either neighboring IR. Some aggregate peaks corresponded to multiple IRs. **c** Assignment of replicated tracks to the nearest IR revealed a triangle centered at each IR, consistent with symmetric replication-fork progression. In contrast to **b**, some tracks extended into a neighboring IR (likely reflecting passive replication). This larger set of replication tracks was used to define IR location: the midpoint of each track was called as an initiation site and IRs were defined as the region between the 25th and 75th percentile of their corresponding initiation sites. Black lines indicate the IR center (median). IR width is displayed on the right y axis. **d** The location of each IR was identified at kilobase scale (median width: 87.5 kb). IRs supported by <5 replication tracks were excluded. **e** About 78.9% of IRs were localized to a region 100 kb or narrower. In the example shown, 59 replication tracks overlapped the IR. The 25th–75th percentile of midpoint locations for these tracks fell within a 60 kb range (dotted lines). **f** Broad IRs may reflect the presence of multiple distinct initiation events that were not disambiguated, technical noise, or mild asymmetry in replication-fork progression. In the example shown, 62 cells overlapped the IR. Visually there appear to be multiple distinct clusters of track midpoints. See Supplementary Fig. 5.

initiation site within the IR was identified. IRs were localized to a median width of 87.5 kb (~4 windows, Fig. 3d), which corresponds to an inter-IR distance of 50–920 kb (median: 260 kb). Most IRs (78.9%) were 100 kb or narrower (e.g., Fig. 3e). Visual inspection of broad IRs (>120 kb) suggested that many contain multiple initiation events that were grouped together because of overlap between replication tracks (Fig. 3f; Supplementary Fig. 5). Thus, while we cannot determine whether IR width (and variability in IR width) reflects technical noise, inconsistency between cells in the precise location of initiation, or mild asymmetry in sister replication-fork progression, we conclude that initiation events are relatively localized and that at least some of the IR widths are likely overestimated. Localized initiation regions are also apparent in the early S fractions of the 10- and 100-fraction profiles (Fig. 1d), where the impacts of noise are averaged across many cells.

In our analysis of IRs, we did find evidence of ectopic replication initiation: only 29.2% of IRs contained a peak in the $S/G_1$ aggregate profile, and 31.5% of IRs were supported by a single replication track. However, these potentially ectopic events comprised a small fraction of all observed initiation events. Rather, 2595 IRs (34.5%) accounted for 90% of all replication tracks, indicating that about a third of the IRs are used consistently across cells. Thus, contrary to previous studies that analyzed single-cell replication profiles at the level of large chromosomal domains[15,16], our data reveal localized initiation regions, which we assume correspond to individual, or tight clusters of, replication origins.

**Consistent yet nondeterministic order of replication initiation.** Given that single cells appear to initiate replication primarily from a consistent set of genomic locations, we turned our focus to the temporal axis of variation: how consistent is the order in which single cells initiate replication at these loci?

We first asked whether the single-cell data were compatible with strictly determined replication timing, such that every cell initiates replication at every IR in the same order. Strict determinism provides a straightforward prediction to test: the number of IRs replicated in any given cell should predict which IRs have been replicated in that cell. For example, a cell that has replicated one IR is predicted to have replicated the IR with the earliest replication timing; a cell that has replicated 100 IRs is predicted to have replicated the 100 IRs with earliest replication timing; and so on. To test how well these predictions matched our data, we counted the number of IRs that were replicated in each cell and used that to assign each IR in that cell an "expected" state —either unreplicated or replicated—assuming that the firing order was fixed (Fig. 4a, b). For a given IR, the observed replication state matched the predicted state in the vast majority of cells (Fig. 4c), indicating that the firing of IRs in single cells follows a highly predictable order. However, we did observe that, on average, an IR differed from its expected state in 11.1% of cells (Fig. 4d). Thus, we can formally rule out the hypothesis that replication timing is strictly determined; IR firing order at the single-cell level is orderly but not entirely predictable.

Having observed variation across cells, we next asked if that variation was uniform across S phase or concentrated at specific times during S phase. We found a parabolic relationship between replication timing of an IR and the proportion of cells that fired that IR out of the strictly determined order. Thus, variability was lowest at the beginning and end of S phase and highest in the middle of S phase, such that 83.4% of the above-average variability occurred in the 53.7% of IRs with aggregate replication timing between 1 and −1. A similar parabolic trend was previously described by Takahashi et al.[15] and was robust in our larger sample size.

We next considered the extent of firing-time variability, asking when in S phase IRs fire in the instances that they fire out of the predicted order. To answer this question, we identified the least-replicated (i.e., earliest) cell in which an IR was observed to fire and the most-replicated (i.e., latest) cell in which it had yet to fire (Fig. 4e). We found that there was an association between the earliest time that an IR fired and its replication timing in the $S/G_1$ aggregate replication profile ($r = -0.64$, Fig. 4f), indicating that IRs with late aggregate timing (hereafter, "ensemble late IRs") tended to start replicating later in S phase than those with early aggregate timing ("ensemble early IRs"). However, most IRs were observed to have fired in a subset of early S-phase cells: 49% of GM12878 IRs fired at least once in a cell with <10% of its genome replicated, 83% in a cell <25% replicated, and 96% in a cell <50% replicated. Thus, many ensemble late IRs were not restricted to firing in late S phase. There was also an association between how late into S phase an IR remained unfired and its aggregate replication timing ($r = -0.66$, Fig. 4g). Thus, ensemble early IRs tended to finish firing across all cells relatively early in S phase, while ensemble late IRs tended to remain unfired into late S phase.

After determining these earliest and latest cells for each IR, we considered them in a paired manner to determine the range of firing times of each IR (Fig. 4h). Given that range is sensitive to outliers (i.e., a duplication called as "replicated" or a deletion called as "unreplicated"), we focused on IRs for which the minimum and maximum values were "corroborated" by a second cell within 10% of S phase from the extreme. Ensemble early IRs tended to first fire in early S phase and to complete their replication before the genome was 50% replicated. In contrast, ensemble late IRs tended to also first fire in early S phase, but to remain unfired in some cells until the end of S phase. Therefore, the firing time of ensemble early IRs was constrained to early S phase, while ensemble late IRs appeared to be less constrained. However, we did observe a small number of IRs that fired exclusively in late S phase across cells; these had a more constrained range. We thus proceeded to further analyze these different behaviors in regions with late aggregate replication timing.

**Ensemble late-initiation regions comprise multiple subtypes.** Our analysis of single-cell replication timing indicated that IRs are fired in a consistent order across most cells, but that ensemble late IRs fire across a larger portion of S phase relative to ensemble early IRs (Fig. 4h). We further dissected the nature of these IRs with large firing ranges to better understand whether we were capturing rare occasions of extremely premature firing or perhaps observing a capacity of IRs to fire throughout S phase. In other words: do these IRs fire substantially ahead of schedule in some cells, or do they not have a scheduled time to fire at all?

We found that each of these two explanations for a large range of firing times were supported by a substantial fraction of IRs, and that neither behavior was sufficient to explain all cases on its own (Fig. 5a). This indicates that some ensemble late IRs tend to fire late but sometimes fire very early, while others fire at many different times in S phase. Specifically, 12% of IRs (27.3% of ensemble late IRs) fired inconsistently throughout S phase (Fig. 5b, e), with earlier aggregate timing corresponding to more cells firing the IR (compare Fig. 5b top left vs. top right). On the other hand, 27% of IRs (63.7% of ensemble late IRs) fired rarely and almost all the replication tracks associated with these IRs were from cells <50% replicated (Fig. 5c, e). Finally, 4% of IRs (9.0% of ensemble late IRs) were never observed to fire in a cell <50% replicated (Fig. 5d, e). Comparing these three classes, IRs that fired throughout S phase tended to have the earliest aggregate

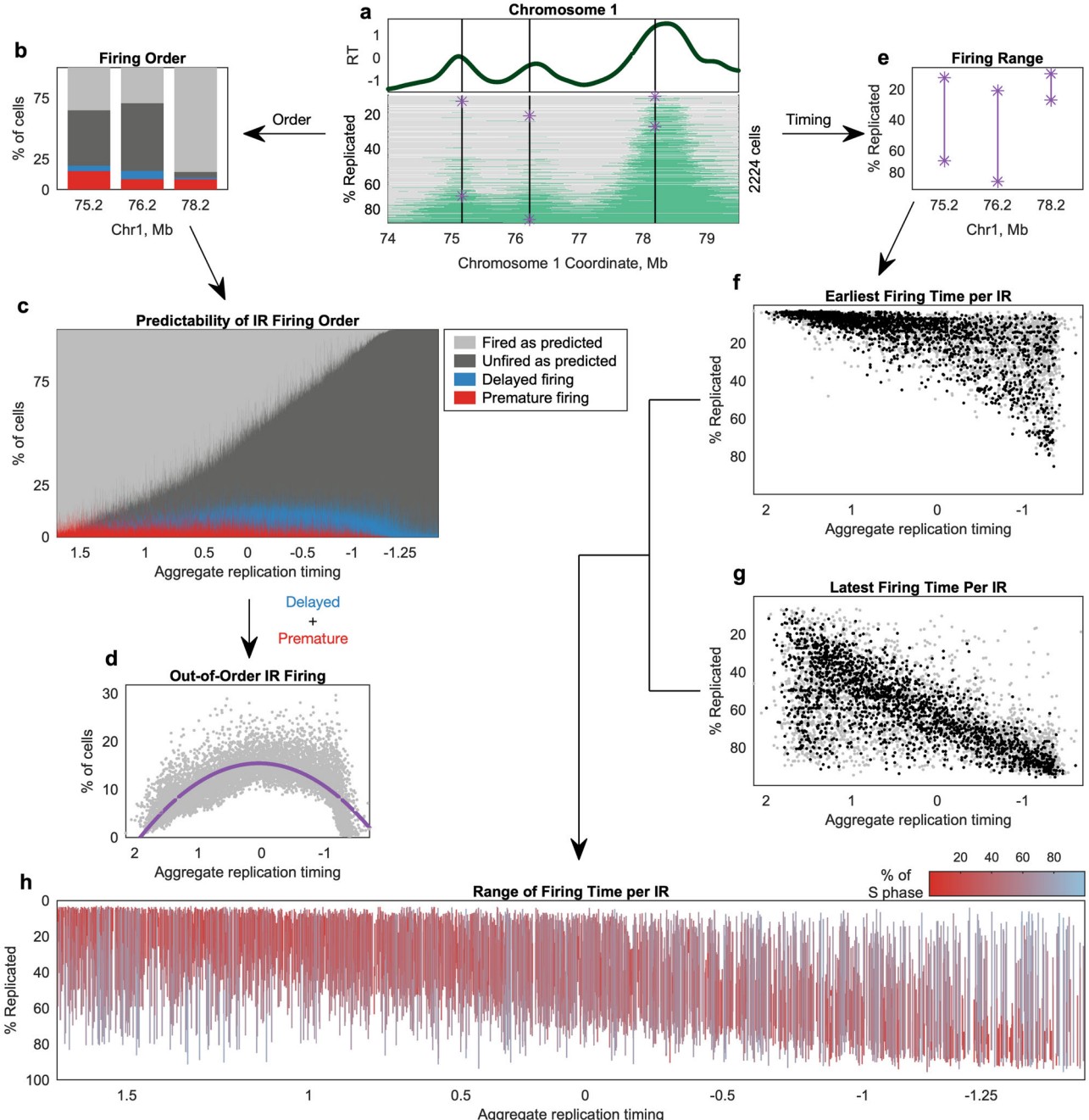

**Fig. 4 Variation in the order and timing of replication initiation in single cells across S phase. a** An example region illustrates analyses of IR firing order (**b–d**) and firing time (**e–h**). Black lines indicate the three IRs in **b** and **e**. Purple asterisks indicate the earliest and latest observed firing times. **b** IRs differ in their degree of consistency across cells. IRs were ranked from earliest to latest, allowing prediction of their replication state under a strict firing ordering. Cells that have replicated an IR not predicted to fire are considered "premature" (red), while those that have not replicated an IR predicted to have fired already are "delayed" (blue). **c** IRs are fired in the expected order in most cells. Columns represent IRs, ranked from earliest to latest (left-to-right). **d** IR firing varies most for IRs expected to fire in mid-S phase. On average, IRs behaved differently than expected in 11.1% of cells (range: 0.3–29.7%). Dots represent IRs. The purple line shows a second-order polynomial fit. **e** IR firing range spanned from the earliest observed fired cell to the latest observed unfired cell. S-phase progression was measured as percent of the genome replicated. **f**, **g** For each IR, we identified the least-replicated and most-replicated cell containing a replication track assigned to that IR. Dots represent IRs: black dots indicate IRs for which a second cell within 10% S-phase progression "corroborated" the earliest and latest firing time. **h** Aggregate early IRs tended to have narrower ranges of firing times than aggregate late IRs. Vertical lines represent IR firing range for IRs with corroborated values, color-coded by the % of S phase during which that IR fires (i.e., the length of the line). A small number of constitutively late IRs (short red lines with late aggregate replication timing) can be observed.

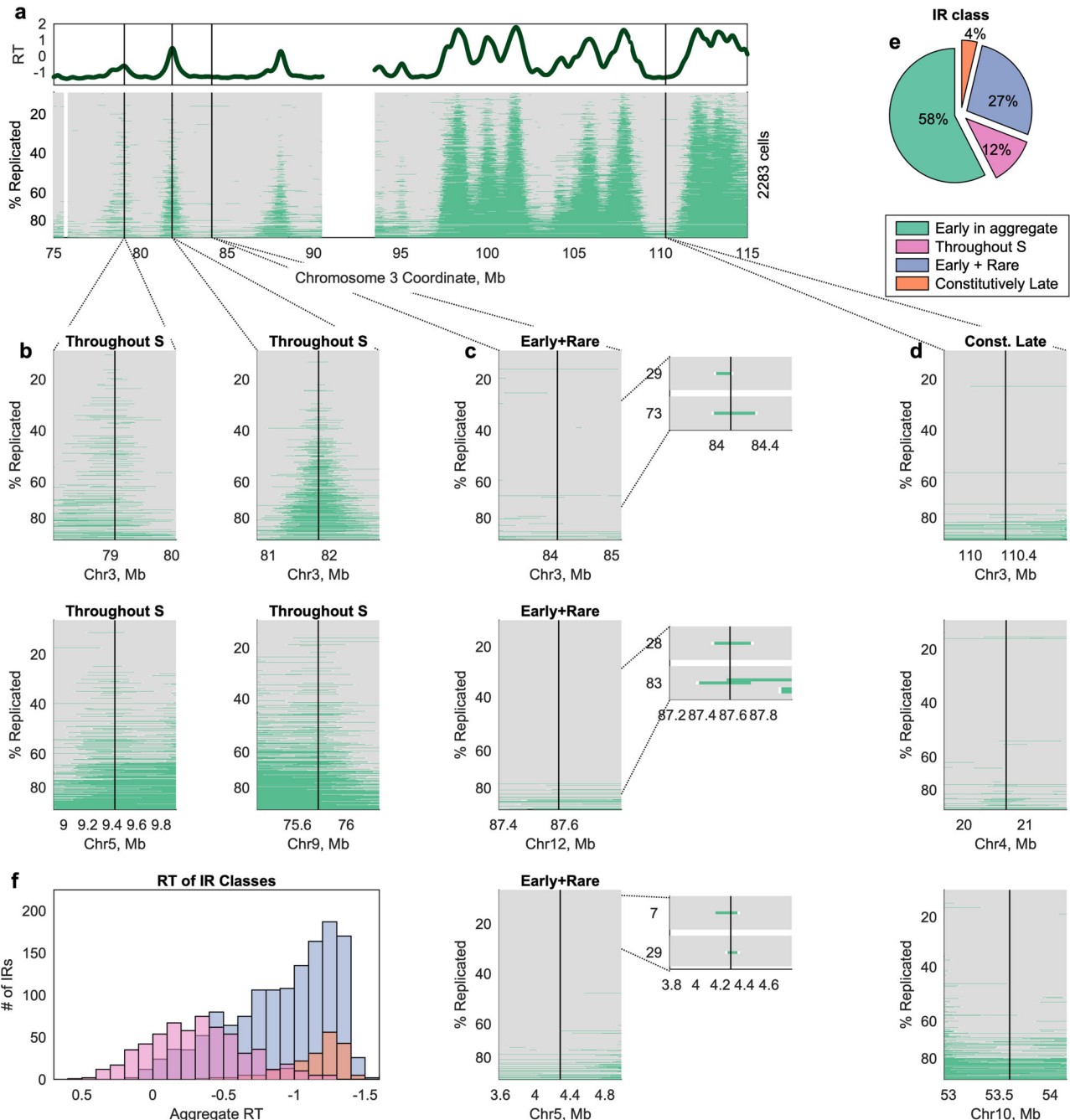

**Fig. 5 Three distinct classes of IRs with late aggregate replication timing. a–d** Ensemble late IRs can be classified into three classes based on their behavior across single cells: some fire throughout S phase (**b**), some fire rarely but often fire early when they do fire (**c**), and some were never observed to fire early (**d**). The IRs indicated with black lines in **a** are shown in the top row of **b**, **c**, and **d**. Additional examples are shown below. **e** About 27% of IRs with late aggregate replication timing fire infrequently but with a preference for early S phase, while 12% fire throughout S phase. Constitutive late firing is rare (4% of IRs). **f** IRs that fire throughout S phase (pink) tend to have earlier replication timing than the other two classes of IRs, while those that were constitutively late (orange) had the latest average replication timing.

replication timing (median: −0.32 and as early as 0.54), while constitutively late IRs had the latest aggregate timing (median: −1.22, Fig. 5f). These unexpected results demonstrate that the late-replicating regions observed in ensemble assays contain origins with heterogeneous firing behavior; these results cannot be fully explained by either a deterministic timing model (which posits these regions contain constitutively late-firing origins) or existing stochastic firing models (which posit that these regions contain low-efficiency origins that become increasingly likely to fire as S phase progresses[11]).

**Single-cell replication timing across cell lines throughout S phase**. Having established a workflow for high-throughput replication analysis of unsorted cells, we performed whole-genome sequencing of 9658 single cells across eight additional cell lines: two LCLs, three embryonic stem cell lines (ESCs), and three cancer cell lines. As with GM12878, we performed in silico cell sorting to distinguish replicating and nonreplicating cells within each library (Supplementary Fig. 6a). For each cell line, we generated an aggregate S/G₁ profile that was highly correlated to an S/G₁ bulk replication-timing profile for the same cell line

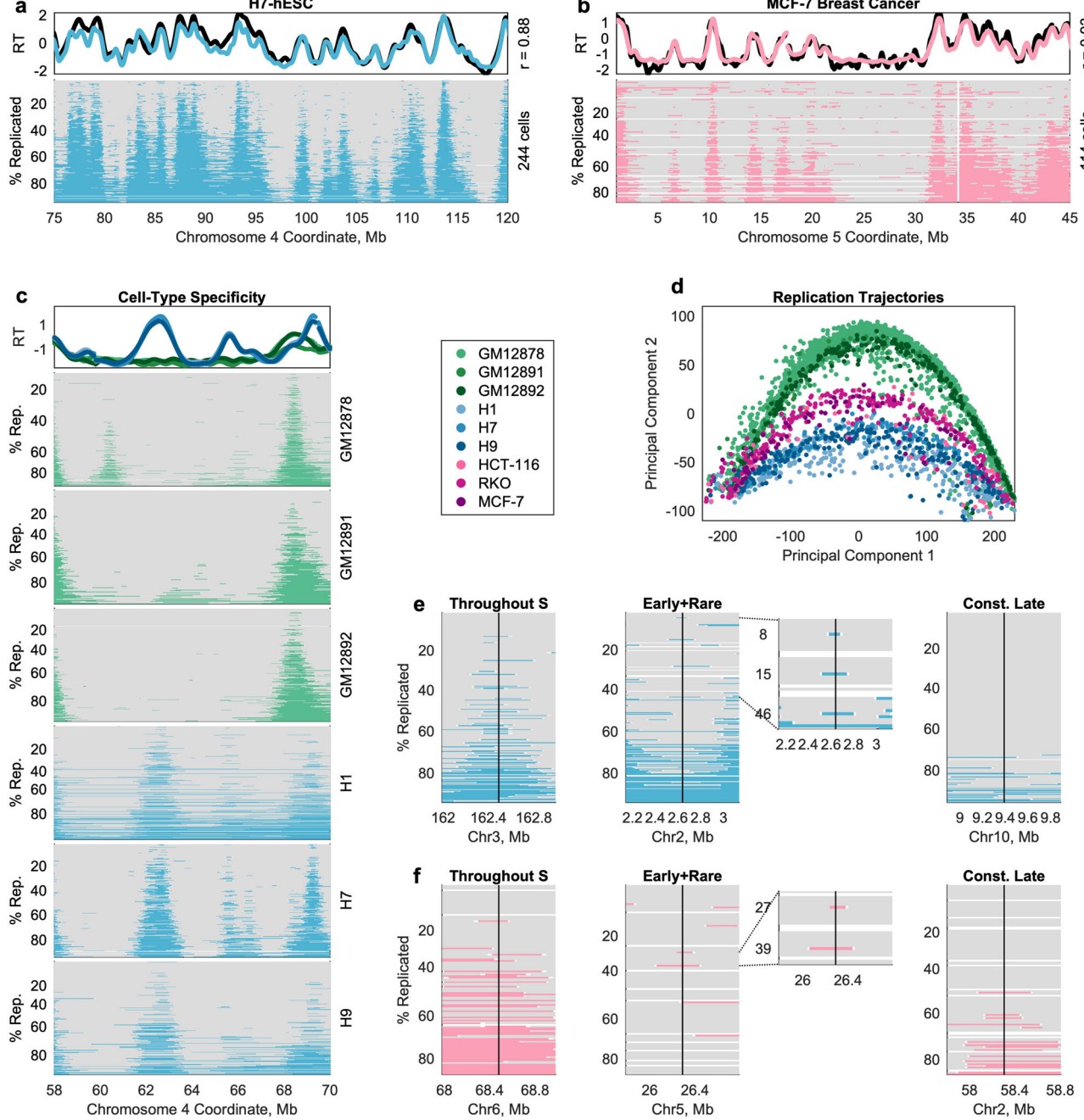

**Fig. 6 Comprehensive measurement of single-cell replication timing across cell types. a, b** As in Fig. 2a, for the embryonic stem cell line H7 (**a**) and for the breast cancer cell line MCF-7 (**b**). **c** Replication-timing variation between cell types is observed at the single-cell level. *Top panel*: bulk-sequencing consensus replication-timing profiles for LCL (green) and hESC (blue). *Lower panels*: single-cell data reveal that the bulk-sequencing peaks at ~62 Mb and ~65.5 Mb reflect the presence of hESC-specific initiation sites. **d** Single cells follow cell-type-specific trajectories of S-phase progression, as determined by principal component analysis (PCA). PCA was performed on replication states in all genomic windows across autosomes. PC1 corresponds to the % of the genome replicated ($r = 0.99$), with negative values of PC1 reflecting early S phase and positive values reflecting late S phase. Cell types segregate along PC2. Each dot represents a single cell. **e, f** All three categories of IRs with late aggregate replication timing described in Fig. 5 were also observed in H7 (**e**) and MCF-7 (**f**).

($r = 0.84$–0.97, Supplementary Fig. 7). We then generated replication profiles for between 110 and 501 S-phase cells across the different cell lines (Fig. 6a, b; Supplementary Fig. 7).

The aneuploid breast cancer cell line MCF-7 highlights the broader applicability of in silico sorting. While we apply this method to focusing our analysis only on replicating cells, it is also valuable in single-cell analysis of copy-number aberrations (CNAs) in cancer. In that context, it is necessary to remove replicating cells prior to CNA calling, since both replication and duplications/deletions affect copy-number estimation. MAPD has previously been used to filter out noisy cells in this type of analysis[23]. However, aneuploidy inflates MAPD values (Supplementary Fig. 6a, compare MCF-7 to other cell lines), making it difficult to effectively set a threshold for filtering. In contrast, explicit modeling of $G_1/G_2$ and S cell populations with distinct linear relationships between read coverage and MAPD efficiently

discriminates cells of interest (either for replication analysis or CNA analysis, Supplementary Fig. 6b).

It has been well demonstrated in ensemble experiments that cell types have distinct replication-timing programs, which are shared by cell lines of the same cell type[3,31,32]. Thus, we asked whether cell-type differences among these nine cell lines were preserved at the single-cell level. Indeed, cell-type differences among the aggregate replication-timing profiles were found to be consistent at the single-cell level (Fig. 6c; Supplementary Fig. 8). These differences in replication state between cell types were sufficient to cluster single cells by cell line and cell type (Fig. 6d), suggesting that individual cells of the same cell type follow a similar trajectory through S phase. Two types of replication-timing differences can be observed at the ensemble level: differences in peak locations (i.e., in the location of fired origins) and differences in peak amplitude (i.e., in the timing at which a shared origin is fired). We observe both of these classes of variation at the single-cell level: cell-type-specific peaks in the S/$G_1$ aggregate profile that reflect the presence of a cell-type-specific initiation site (e.g., Supplementary Fig. 8a, right) and peaks of different amplitude in the S/$G_1$ aggregate that correspond to early vs. late firing of a shared initiation site (e.g., Supplementary Fig. 8b, left). Most intriguingly, we also observe a novel type of cell-type difference invisible to ensemble profiling methods: a subset of cell-type differences that appears to be driven by inconsistent usage of an initiation site in one cell type (e.g., Supplementary Fig. 8a, left ~196.1 Mb).

We proceeded to call IRs in each cell line and repeated the above analyses of IR order and timing variability. Despite having ~10 times fewer cells relative to GM12878, we were able to identify 1811–5055 IRs (compared with 7522 in GM12878) per cell line in all cell lines, except for HCT-116 (discussed below). To directly test the hypothesis that we identified fewer IRs because of the smaller number of cells, we performed downsampling of the GM12878 cell line and confirmed that the number of IR calls rapidly increases with increasing sample size (Supplementary Fig. 9). IRs called for other cell lines were slightly broader than the GM12878 IRs, but still localized (median: 110–220kb, Supplementary Fig. 10a). This suggests that increasing the number of cells analyzed will likely yield additional IRs in all cell lines and also further narrow their localization.

Patterns of initiation-site localization and timing variability across cell lines were broadly consistent with those observed in GM12878, even though the specific locations of IRs differed between cell types. IRs were fired in a predictable but not fixed order (Supplementary Fig. 10b) that was more disordered for those IRs with mid-S-phase aggregate timing (Supplementary Fig. 10c). With regard to firing time, ensemble late IRs fired early in S phase in a subset of cells, although with the smaller sample size, fewer IRs had multiple cells corroborating this behavior (Supplementary Fig. 11a, b). This is consistent with how rarely these events occurred per IR in GM12878 and suggests that these events would be observed more frequently in other cell lines when looking across a larger number of cells. However, the fact that so many rare events are observed even in a sample size of ~200 cells suggests that the full scope of variability remains underestimated, including in GM12878.

Finally, the three classes of ensemble late IRs were present in each cell line (except HCT-116), and two features were common between GM12878 and other cell lines. First, rarely-used IRs with a preference for early firing were more common than IRs that fired throughout S phase; second, a small fraction (4–11% in most cell lines) of IRs were constitutively late (Fig. 6e, f; Supplementary Fig. 11c).

The outlier cell line was the colorectal cancer cell line HCT-116, for which we recovered only 110 replicating cells and

identified only 768 IRs. In addition to wider IRs, with a median width of 280 kb, 78% of IRs identified in HCT-116 were ensemble early IRs. (In other cell lines, this value was close to 50%, in line with the genome-wide replication-timing values.) This bias toward discovering IRs in early-replicating regions creates the impression that variability increases monotonically across S phase, particularly when examining HCT-116 alone. These results are presented alongside those of the other cell lines to illustrate how a low cell count can bias IR identification and conclusions drawn from subsequent analyses. However, while not particularly informative about IRs in late-replicating regions, the data from HCT-116 are not incompatible with the trends observed for early IRs across cell lines.

In addition, we repeated all analyses for the DLP + GM18507 library. Single-cell replication profiles for this cell line were noisier (Fig. 2b), and we called 12,952 IRs from 759 cells. GM18507 IRs recapitulated the results from the 10x Genomics cell lines, with the caveat that these cells were strongly skewed toward mid-S phase (Supplementary Fig. 10; Supplementary Fig. 11).

In summary, data from ten human cell lines encompassing LCLs, ESCs, and cancer cell lines support the conclusion that replication initiation occurs in localized regions that are largely consistent across cells. Furthermore, patterns of heterogeneity in origin firing order and firing time appear to be generalizable across cell lines and cell types.

## Discussion

While ensemble replication-profiling methods cannot capture (and may be confounded by) cell-to-cell heterogeneity, previous single-molecule and single-cell methods have been largely limited in their throughput or accuracy. Here, we report a scalable method for analysis of thousands of single replicating cells, across multiple cell lines, and at kilobase resolution. We describe an in silico strategy to sort cells across S phase, analogous to and more accurate than traditional flow cytometry, and demonstrate how this method enables simultaneous analysis of replication initiation at population, subpopulation, and single-cell resolutions. In addition, by focusing specifically on replication initiation events called from single cells, we are able to identify which cells are informative about which replication initiation sites, capturing information that is analogous to that collected from lower-throughput single-molecule studies. In a parallel study, Gnan et al.[33] developed a similar approach to use single-cell sequencing data to infer DNA replication timing at large scale.

We find that single cells primarily initiate replication at consistent loci, corresponding to peaks in the replication-timing profile. Across cells, we are able to pinpoint the locations of 78.9% of these initiation events to regions no larger than 100 kb (likely overestimated due to low coverage), challenging the model that there are megabase-long replication domains that are replicated simultaneously[5,6]. Analogously, our data do not support the existence of large constant replication regions (CTRs)[34], particularly in early-replicating regions, for which we have more data. While it is conceivably straightforward to envision how measurements with limited resolution would give the impression of domains or CTRs where none exist, it appears more difficult to reconcile the sharp and discrete initiation peaks in our single-cell data with the idea of large regions with constant replication timing. In contrast, our data are consistent with recent high-resolution studies that suggest that replication initiation is confined to regions of several tens of kilobases[7,10,30,35]. Our observation that even tight peaks in ensemble replication-timing profiles often encompass multiple discrete single-cell initiation events lends further credence to the argument that initiation

events are even more localized than measured here, with the caveat that we cannot distinguish in our data between single origins and tight clusters of nearby origins. We find evidence for ectopic initiation from regions outside these commonly used initiation regions (as in[7]), although these events comprise a small fraction of overall events. While many previous studies of mammalian replication origins relied on biochemical enrichments of DNA-synthesis events and are therefore more prone to false-positive identification of apparent initiation events, single-cell DNA sequencing more reliably represents productive and internally validated DNA replication[1].

While spatial variability in replication initiation is rare, temporal variation is more common. In general, initiation regions (IRs) expected to fire in the middle of S phase are more variable than those expected to fire earlier or later, consistent with previous reports[15]. At the level of individual IRs, we find that many, particularly those with early aggregate replication timing, have a preferred time of firing that is captured by the aggregate replication-timing profile. IRs with early aggregate replication timing tend to be fired in all cells early in S phase, while those with late aggregate replication timing fire across a broader range of S phase. We further find that late-replicating IRs can be divided into multiple classes, with only a small subset (<10%) firing constitutively late. Instead, most late IRs can and do fire early—sometimes rarely and sometimes often.

Our data do not rule out the possibility of a global regulator (or regulators) that dictates replication timing in a semideterministic manner. However, they are also consistent with the more parsimonious model that origin-specific firing probabilities produce a relatively consistent replication-timing landscape in single cells. IRs with late aggregate replication timing that occasionally fire early in S phase are consistent with this hypothesis: these rarely early IRs could contain an inefficient origin (or clusters of inefficient origins) that rarely fires but can be early firing when it does fire, and this low efficiency is what is measured by the aggregate replication-timing profile. The constitutively late IRs have even later aggregate replication timing; under this same hypothesis, they would be expected to fire early in S phase even less often. Thus, we cannot rule out the possibility that the IRs we observed to be constitutively late do sometimes fire early, but at so low a frequency that this behavior was not captured in our sample.

While our data are consistent with an important role for origin firing efficiency in determining replication timing, the distinct classes of initiation regions we describe also highlight a shortcoming of considering origin efficiency at the level of individual loci: while low-efficiency origins would be expected to rarely fire in early S phase, their probability of firing should remain constant or even increase as S phase progresses[11]. In other words, origins in late-replicating regions of the genome should fire throughout S phase. Instead, we see that the majority (63.7%) of the inefficient IRs have a low probability of firing in early S phase, and a negligible probability of firing later in S phase (Fig. 5e), suggesting that the context of replication initiation changes across S phase in a manner that has not been previously characterized.

Our results suggest that origin-specific firing efficiencies play a key role in producing the replication-timing program; as such, they underscore the value of future work parsing out the contributions of DNA sequence, gene expression, chromatin accessibility, and doubtless other factors to these firing efficiencies. At the same time, a future model for replication timing must also explain why many origins appear to have their highest probability of firing at the beginning of S phase, rather than becoming increasingly likely to fire as S phase progresses—and also why that does not result in large regions of underreplication that persist into $G_2$ phase, as modeled in[11].

While single-cell sequencing provides insight into cell-to-cell variability that ensemble measurements cannot capture, such experiments are also more expensive, more time-consuming, and require more complex analysis methods. The approach we describe here is limited by the high cost of existing commercial microfluidics-based library-preparation methods, especially given that not every cell will be informative about every locus. Thus, there is a trade-off between sample size and information content per cell. In this study, we have demonstrated that low per-cell sequencing coverage is sufficient for distinguishing the twofold copy-number difference between replicated and unreplicated regions at ~20-kb resolution. However, we are unable to reliably distinguish smaller differences in copy number, i.e., between 2 and 3, or between 3 and 4 copies. Increased resolution and/or allele-specific mapping[15,16] would be required to identify cases of allelic asynchrony, especially those specific to individual cells. In addition, single-cell sequencing is, at least currently, not the optimal technology for identifying individual replication origins, and existing origin-mapping methods (e.g., OK-seq[30], EdUseq-HU[35], high-resolution Repli-seq[10], or optical replication mapping[7]) are better suited to this purpose. Origins identified by these methods overlap well with ensemble replication-timing profiles, as do the IRs we identified here.

Single-cell DNA sequencing of proliferating cell samples, without experimental manipulation (e.g., cell synchronization or sorting), can reveal the dynamics of DNA replication in exquisite detail. Applying this approach across cell types, genetic backgrounds, and experimental conditions will reveal how replication is altered at the spatiotemporal level in different physiological contexts. With constantly improving methods for high-throughput single-cell isolation and accurate whole-genome amplification[19,36,37], this approach promises to become ever more informative for the understanding of the DNA replication timing program.

## Methods

**Cell culture**. Lymphoblastoid cell lines (GM12878, GM12891, and GM12892) were obtained from the Coriell Institute for Medical Research and cultured in Roswell Park Memorial Institute 1640 medium (Corning Life Sciences, Tewksbury, MA, USA), supplemented with 15% fetal bovine serum (FBS, Corning). Embryonic stem cell lines (H1, H7, and H9) were obtained from the WiCell Research Institute (Madison, WI, USA) and cultured feeder-free on Matrigel culture matrix in mTeSR™ 1 medium (WiCell). Tumor-derived cell lines (MCF-7, RKO, and HCT-116) were obtained from the American Type Culture Collection. MCF-7 and RKO cells were cultured in Eagle's minimum essential medium (Corning), supplemented with 10% FBS. HCT-116 cells were cultured in McCoy's 5a medium (Corning), supplemented with 10% FBS. All cell lines were grown at 37 °C in a 5% $CO_2$ atmosphere.

**Library preparation and sequencing**. For sorted libraries, GM12878 cells were stained with Vybrant™ DyeCycle™ Green Stain (ThermoFisher Scientific, Waltham, MA, USA) and sorted into five fractions ($G_1$-, $G_2$-, early S-, late S-, and full S-phase) with a BD FACSMelody™ Cell Sorter (BD Biosciences, Franklin Lakes, NJ, USA).

For both sorted and unsorted libraries, isolation, barcoding, and amplification of single-cell genomic DNA was performed on the 10x Genomics Chromium Controller instrument, using the 10x Genomics Single Cell CNV Solution kit (10x Genomics, Pleasanton, CA, USA). Paired-end sequencing was performed for 100 cycles with the Illumina NovaSeq 6000 (10x Genomics), 150 cycles with the Illumina HiSeq X Ten (GENEWIZ, Inc., South Plainfield, NJ, USA), or 36 or 75 cycles with the Illumina NextSeq 500 (Cornell University Biotechnology Resource Center, Ithaca, NY, USA). For libraries sequenced multiple times, FASTQ files were merged prior to downstream processing. See Table S1 for details.

**Processing of single-cell barcodes**. Single-cell barcodes were processed with a custom python script. Specifically, the first 16 bp of each R1 read (containing the cell-specific barcode) was trimmed with seqtk (v1.2-r102-dirty). Raw barcode sequences were compared with a whitelist of 737,280 sequences (10x Genomics) and filtered by abundance to produce a list of barcodes present in the library. Specifically, a set of high-count barcodes was identified as those that were represented at least 1/10 as often as the highest-abundance barcode. A minimum

barcode-abundance threshold was then set as 1/10 the 95th percentile of the high-count abundances.

Next, we attempted to correct barcode reads that were not found in the set of valid barcodes. To be corrected, we required that the barcode read contain no more than one base position with a quality score <24 and that there was only one valid barcode with a Hamming distance of 1.

**Processing of sequencing reads**. After filtering out sequencing reads without a valid barcode, reads were aligned to the human reference genome hg37 (humanG1Kv37) using the Burrows–Wheeler maximal exact matches (BWA-MEM) algorithm (bwa v0.7.13). Barcodes were then merged into the aligned BAM files using a custom awk script, and barcode-aware duplicate marking was performed using Picard Tools (v2.9.0). High-quality (MAPQ ≥ 30) primary mate-pair alignments were included in further analysis. Members of a mate pair were counted together if they were mapped within 20 kb of one another (weight of 0.5/read), and separately (weight of 1/read) if not. After read alignment, all downstream analyses were performed in MATLAB (v2019a).

**Identification of $G_1/G_2$ cells and definition of $G_1$ windows**. Reads were counted in fixed-size windows of 20 kb. After removing low-mappability windows (in which fewer <75% of nucleotide positions were uniquely mappable[38]), sets of 50 windows were aggregated together to calculate the median absolute deviation of pairwise differences between adjacent windows (MAPD)[20]. MAPD was then divided by the square root of the mean number of reads per aggregated window (mean coverage/Mb), to produce a linear relationship between coverage and scaled MAPD. For each sequencing library, an expectation-maximization procedure was used to fit the data as a mixture of two Gaussian functions. The linear fit predicting the smaller scaled MAPD value at the maximum observed coverage was assumed to model the $G_1/G_2$ relationship between coverage and scaled MAPD, and cells with a residual ≤0.05 from this model were assigned as $G_1/G_2$.

Next, we defined a set of variable-size, fixed-coverage windows using a $G_1$ control, along the lines of Koren et al.[22]. In this case, the $G_1$ control was created in silico by aggregating reads from a subset of $G_1/G_2$ cells, prioritizing high-coverage $G_1/G_2$ cells. (The number of cells used varied between libraries and was determined as the number of cells that would define windows of ~20 kb.) This was performed independently for each sequencing library prepared from the same cell line. Per-cell read counts were calculated in these $G_1$ windows, to account for mappability and GC-content bias, as well as any copy-number variations that were common to many cells within the library.

Finally, we identified and filtered out cell-specific copy-number aberrations (CNA). To do this, we fit a two-component mixed Poisson model to aggregated read counts (15 windows, ~300 kb), and searched for the genomic region with the lowest probability of being observed under either rate coefficient, λ. If the median probability of each window within this region was less than the maximal probability of all windows genome-wide, we determined it to be a CNA and masked the read counts in that region. This process was performed iteratively until no new regions were discovered. Cells with an autocorrelation in read counts >0.15 after filtering were assumed to have residual undetected CNAs and were excluded from analysis.

**Replication-state inference**. For each cell, we assigned each $G_1$-defined window as "replicated" or "unreplicated" using a two-state hidden Markov model (HMM). To initialize the model, we again fit a two-component mixed Poisson model to aggregated read counts (15 windows, ~300 kb) and assigned each window to the mean it was closer to. If this initial model did not converge, or if the ratio between the two mean copy numbers was not ~2 (between 1.5 and 2.5), the cell was excluded. Otherwise, we refined the initial window assignments using the HMM, which modeled read counts as the mixture of two Poisson processes.

Because the HMM does not model the expected twofold relationship between replicated and unreplicated regions, we assessed the quality of the HMM output using this ratio. Specifically, we calculated the ratio between the average number of reads in windows assigned as replicated to the average number of reads in windows assigned as unreplicated. To be included in further analysis, this ratio was required to be between 1.5 and 2.5. This filter removed cells poorly modeled by the HMM, which could be explained by a variety of biological and technical factors, including large CNAs, large replication defects, ineffective selection of cell-specific initialization parameters, and atypical noise.

In addition, to find any cells that contained uncorrected CNAs, we performed three filtering steps. First, we calculated the average copy number assigned to each chromosome and excluded cells for which the standard deviation between chromosomes was greater than 0.4. Second, any cell that contained both a fully unreplicated chromosome and a fully replicated chromosome was excluded. Third, we calculated the pairwise correlations between cells for each chromosome individually. If the mean pairwise correlation between a cell and all other cells was negative, or if the pairwise correlation between a cell and one of its 10 closest neighbors was a statistical outlier, that chromosome was excluded for that cell.

Finally, for the ease of analysis, we interpolated the data back onto fixed-size 20 kb windows. Interpolated values for fixed-size windows that overlapped multiple $G_1$-defined windows were not always integers. Thus, windows assigned a noninteger copy number were masked, as were low-mappability windows.

**Assessment of HMM resolution**. To assess the resolution of HMM copy-number inferences, read counts were simulated for 2500 single GM12878 cells following a strictly determined replication-timing program. First, the bulk replication-timing profile was divided into 1000 equally spaced bins (corresponding to ~0.046 replication-timing units). For each vertical "time point" slice through the bulk profile, windows with earlier replication timing were assigned "4N" and windows with later replication timing were assigned "2N". For each simulated cell, one of these 1000 possible time points was randomly selected, and an average coverage was drawn from the distribution of observed coverage values for the unsorted GM12878 library. Then, read counts for 2N and 4N windows were drawn from two Poisson distributions, with rate coefficients selected to produce the desired average coverage. Finally, the simulated read counts were run through the replication-state inference pipeline.

**Bulk-sequencing replication-timing profiles**. Replication-timing profiles from bulk-sequencing assays were used to benchmark single-cell replication profiles. For GM18507, an LCL consensus profile[22] was used. For all other cell lines, a profile for the specific cell line was used. For Illumina Platinum LCLs (GM12878, GM12891, and GM12892)[39] and hESCs (H1, H7, and H9)[8], these data are previously published.

**Aggregate replication-timing profiles**. For each cell line, we generated an aggregate S/$G_1$ profile, as in[22], except that we generated the $G_1$ and S fractions in silico by aggregating reads across all cells assigned to that fraction. Briefly, the $G_1$ fraction was used to generate variable-size windows with a fixed number of reads ($n = 200$), and the number of S-phase reads was then counted in the same windows. This profile was smoothed in a gap-aware fashion with a cubic smoothing spline (MATLAB function csaps), with a smoothing parameter of $10^{-16}$, and normalized to a mean of 0 and standard deviation of 1.

**Sub-S-phase fraction profiles**. To generate a profile for 10 sub-S-phase fractions, we partitioned cells into 10 bins of equal cell population, based on the % of the genome replicated. We summed the read counts (in $G_1$-normalized windows) across all cells within each partition. To normalize read counts between fractions, we then scaled these values, setting the 1st percentile value as 2 and the 99.9th percentile value as 4. The same procedure was used to generate 100 fractions.

**Identification of initiation regions**. To identify single-cell replication-initiation sites, we began by defining all replicated segments ("replication tracks") across the genome of each cell. These segments were defined as contiguous windows with inferred copy number of 4, containing no more than 5 consecutive masked windows. As a first approximation of the locations of replication initiation, the midpoint of each replication track was assigned as the most likely site of initiation. (Replication tracks longer than 1 Mb were excluded from this initial analysis.)

To cluster single-cell initiation sites, we grouped together replication tracks that overlapped one another. We considered three possible midpoints for each replication track: the observed midpoint as well as the midpoint if either the left or right boundary had been misplaced by 2.5 windows. Starting with the shortest replication tracks, we asked whether each replication track overlapped any previously defined initiation regions (IRs). Tracks overlapping a single IR were attributed to activity of that IR (as long as its midpoint overlapped at least one track already assigned to that IR), while tracks that did not overlap any IRs were used to define a novel IR. Tracks that overlapped multiple IRs were inferred to reflect the activity of multiple initiation events and were not used to define IRs. This clustering procedure considered only relative replication-track length, and no global threshold value was used to exclude tracks.

After defining IRs, we reconsidered any track less than 1 Mb in length that had not been attributed to an IR (i.e., tracks that overlapped multiple IRs). These tracks were then assigned to the IR closest to its midpoint. The width of each IR was calculated from the 25th percentile to the 75th percentile of the midpoints of replication tracks attributed to the IR, and the center was set at the 50th percentile. IRs supported by fewer than 5 tracks were not included when calculating the median IR width.

**Variation in firing order across cells**. To assess variation in the order in which IRs were fired across cells, we compared the data to a null model under which every cell fires the same IRs in the same order. Under this model, the number of IRs inferred to be replicated also dictates which IRs those are. Thus, we counted the number of replicated regions overlapping IRs in each cell, and then predicted which regions those would be under the null model. For each IR, we then calculated how many cells did not match our prediction.

**Variation in firing time across cells**. To determine the range of firing orders for each IR, we identified the earliest cell containing a replication track attributed to an IR, and the latest cell in which the center of the IR was inferred to be unreplicated (after excluding outlier cells that had not replicated any of the neighboring IRs). The percent of the genome replicated in each of these cells was used as a proxy for

time during S phase. Given that range is a metric extremely sensitive to outliers, we considered an IR's range to be "corroborated" if a second cell was observed within 10% of its earliest and latest firing time. We focused on these IRs with corroborated ranges in subsequent analyses.

Finally, we classified IRs that fired in fewer than 50% of cells into three groups based on their firing behavior throughout S phase. To do this, we considered the percent of the genome replicated in each cell containing a replication track attributed to that IR. IRs that were not associated with any cells <50% replicated were considered constitutively late firing, while those associated with more than 5 cells >50% replicated were considered to fire throughout S phase. The remaining IRs, which were associated with 1–5 cells in early S phase, were considered to be rarely fired with a preference for early firing.

**Reporting summary**. Further information on research design is available in the Nature Research Reporting Summary linked to this article.

## Data availability

Sequencing data generated in this paper were deposited at the Sequence Read Archive (SRA) under accessions PRJNA770772 (single cell) and PRJNA419407 (bulk). Bulk-sequencing replication-timing profiles used for comparison are available at http://www.thekorenlab.org/data. The human reference genome assembly humanG1Kv37 used in this study was downloaded from the International Genome Sample Resource (IGSR, http://www.internationalgenome.org) [http://ftp.1000genomes.ebi.ac.uk/vol1/ftp/technical/reference/human_g1k_v37.fasta.gz]. Source data for the FACS sorting of GM12878 cells are provided with this paper.

## Code availability

All scripts used in data processing, analysis, and visualization are available on GitHub at https://github.com/TheKorenLab/Single-cell-replication-timing (https://doi.org/10.5281/zenodo.6463446)[40].

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

## Acknowledgements

We thank Claudia Catalanotti and Rajiv Bharadwaj for sharing data analyzed in this work, Peter Schweitzer and Jennifer Mosher for technical support and guidance with library preparation and sequencing, Kevin Massey for many fruitful discussions about algorithms, Sneha Sharma for help with various preliminary analyses, and members of the Koren lab for their feedback. This work was supported by the National Institutes of

Health (grant DP2-GM123495 to A.K) and a seed grant from the Cornell Center for Vertebrate Genomics (A.K.).

## Author contributions

D.J.M. and A.K. conceptualized the project. D.J.M. performed experiments and analyses. D.J.M. and A.K. wrote the paper.

## Competing interests

The authors declare no competing interests.
