## [Peer Review File · Nature Communications]

REVIEWER COMMENTS

Reviewer #1 (Remarks to the Author):

The authors utilized scWGS to study DNA replication profiling, that enables more accurate and more sensitive characterization of replication origin firing (initiation regions), compared to traditional flow cytometry. Using this new approach, the authors confirmed existing knowledge by demonstrating that replication initiation events in single cells are mostly the same as those observed in the population average. But they also identified a subset of initiation regions that are outliers, that fire constitutively throughout the S phase. The new approach and the new scientific findings included in the manuscript are worthwhile for publication and will be of great interest to the relevant community. Before that, several aspects should be improved to make the manuscript more solid scientifically and better in overall quality.

(1) In P4, opposite to the authors' claim, MDA is actually worse than DOP-PCR or DLP for CNV analysis. The key advantage of 10X Genomics platform is its high throughput, enabling the analysis of thousands of single cells simultaneously in one run.

(2) In P6, I agree there are many benefits of using single-cell whole-genome sequencing to profile DNA replication, compared with traditional FACS. But it is still better to list the cons as well as the pros the authors advocated, such as a higher cost and a longer time for experiment procedure and data processing.

(3) In P7, the authors suggested that the symmetric track is due to bidirectional replication fork progression from a single origin. How to rule out the scenario of multiple origins fired in each observed track? The authors came to this point later by choosing <1 Mb tracks, but here when the phenomenon is described in the manuscript for the first time, it is better to clarify a bit.

The major concern is that, because single-cell WGS generate genome-wide noise and bias on top of the DNA replication profile, the authors have to be very careful in data interpretation, especially the analysis leading to the scientific conclusion that "there are two subsets of initiation regions" highlighted in the abstract. Along this line, I have several suggestions as following.

(4) In P4, for the 2.4% of cells in G1 phase by FACS but in S phase by sequencing, and 25.1% of cells in S phase by FACS but in G1 phase by sequencing, the authors seem to be confident to claim that this discrepancy suggest single-cell DNA sequencing provides a better sensitivity. While this likely is true, to be more rigorous, it will be ideal to analyze just these cells with different assignments between FACS and sequencing, and make sure their replication profile and other properties are just the same as other cells with the same assignment from FACS and sequencing. I also wonder if we narrow the FACS gating in S phase (i.e. gating mid-S only), will that reduce the 25.1% number? 25% seems to be such a large number.

(5) In P6, I wonder how robust the analysis result could be, when changing parameters in HMM fitting?

(6) In P9, is the arbitrary threshold of 1 Mb sensitive to the analysis outcome? If change 1 Mb to another number, will the conclusions still hold true? The same applies to later analysis when the authors cluster tracks into IRs by prioritizing shorter tracks over longer tracks. I wonder whether the analysis result is robust when changing such prioritizing parameters.

(7) For DLP+ data, it is good to see it can reproduce the replication timing profile. But will it generate the same conclusions as the authors described later in the paper? Especially the surprising conclusion highlighted in the abstract, that there is a subset of “ensemble late” initiation regions that can fire randomly throughout the S phase.

Reviewer #2 (Remarks to the Author):

Recent studies have demonstrated the power of single cell DNA sequencing for determining their DNA replication program based on the doubled copy number of replicated compared to unreplicated DNA sequences. However, these early studies were limited in resolution and cell number.

Here, Massey and Koren report sequence data from thousands of single cells. They describe an in silico strategy to discriminate S phase cells from other cell cycle stages, which they suggest is more accurate than traditional flow cytometry, and simplifies experimental procedures. They find that single cells initiate replication at consistent initiation regions (IR) < 100 kb. In contrast, the firing time of IR is largely variable from cell to cell. IR that on aggregate fire early in S phase tend to fire early in S phase in all cells, but IR that on aggregate fire late in S phase can with some probability fire at any stage in S phase. Only 10% of late IR appear to fire constitutively late. Finally, by analysing different cell types with principal component analysis, they show that single cells follow cell-type specific trajectories of S-phase progression.

Overall this is a technically strong and well-written story that advances the state-of-the-art of single-cell replication timing analysis. However, the gain in biological knowledge is at best marginal. Several studies, both at the cell population and single molecule or single cell level, have previously reported the locations and replication time distributions of initiation regions in human cells and have proposed based on these data that replication time largely emerges from the stochastic firing of origins with different firing probabilities. While confirmation at the single-cell level is nevertheless important, no effort has been presented here to evaluate the consistency of e.g. IR location with previous IR mapping studies. Furthermore, some of their conclusions seem to be incorrect, as explained below, and several Figures are insufficiently legended.

Major criticisms

1. The studied cell lines are diploid. Unless origin firing is strictly coordinated between the two homologous chromosomes of a pair, one expect at any time in S phase a significant fraction of genomic segments to be replicated on one homolog but not on the other one (copy number = 3N). However, replication profiles were inferred for each cell using a Hidden Markow Model that only considers two possible states: both alleles are unreplicated (2N) or both are replicated (4N). Why did not they run a three-state HMM rather than a two-state HMM ?

Furthermore, in the Methods paragraph "Replication state inference", they mention that "If this initial model did not converge, or if the ratio between the two mean copy numbers was not ~ 2 (between 1.5 and 2.5), the cell was excluded" and that "we interpolated the data back onto fixed size 20kb windows. Windows for which a value other than 2 or 4 was interpolated were masked". Could it be that these filters specifically eliminate cells and/or regions with significant replication asynchrony between homologues ?

2. More detailed comparison of their initiation regions (IRs) with those seen by other techniques (OK-seq, Petryk et al 2016; EdUseq-HU, Tubbs et al 2018; high-resolution Repli-seq, Zhao et al 2020; Optical Replication Mapping, Wang et al 2021) is required. In particular, since they identified 7482 IRs in GM12878 cells, which are very closely related to GM06990 cells used in OK-seq experiments, a direct comparison is feasible. It is not clear whether their analysis brings any novel information about initiation regions.

3. The identified number of initiation events is problematic. It is generally accepted that some 20,000 - 50,000 initiation events take place during each replication cycle in human cells. However, they claim (line 320) that " 2,640 IRs (35.3%) accounted for 90% of the replication tracks, indicating that about a third of the IRs are responsible for the vast majority of initiation events genome wide". If so, only 3,000 initiation events would take place per S phase per haploid genome, and the average spacing between initiation events would be about 1 Mb, which is one order of magnitude larger than repeatedly observed by DNA fiber techniques. In fact, given a fork speed of 2 kb/min, if initiation events were regularly spaced every Mb, it would take a single fork 8h to travel and converge with a fork emitted from a neighboring origin. These kinetics are inconsistent with many studies over the past decades. Clearly, the authors are underestimating the number of initiation events per S phase by a factor of 10. Why ? Can they estimate the apparent speed of replication progression from their pile-up plots and compare with known fork speeds ?

They suggest (line 515) that "increasing the number of cells analyzed will likely yield additional IRs in all cell lines and also further narrow their localization". If so, it is likely that they only detected a fraction of all initiation events in the 2,428 GM12878 cells. Could they subsample this and detect origins at increasing numbers of cells ? If origin number does not saturate at 2,400 cells, origin description cannot yet be comprehensive. If on the other hand origin number is approaching a plateau of 3,000 initiation events, then resolution of the sequencing data is probably the limiting factor in origin detection and increasing the coverage is required before a definitive description can be proposed.

4. I was not entirely convinced that their in silico sorting method performs better than FACS for several reasons.

a) To computationally distinguish replicating cells from non-replicating cells, they quantify local read depth fluctuations using a metric called MAPD (median absolute deviation of pairwise differences between adjacent genomic windows), of which they say nothing for the unfamiliar reader. Please give a reference. In particular, it remained unclear to me why "scaling" (do they mean multiplying or dividing?) the MAPD by the square root of read depth produced a linear relationship between coverage and scaled MAPD. I would have expected that the scaled MAPD would no longer vary with coverage, at least for G1 cells.

b) The choice of window size is critical for using the MAPD to distinguish between replicating and non-replicating cells. The chosen window size was 1 Mb. The discriminatory power should therefore be maximum when replicated and/or unreplicated segments alternate at a characteristic length of 1 Mb. This is probably the case in mid S phase. At the beginning or end of S phase, however, the characteristic lengths of replicated and unreplicated segments may be very different, so that most pairs of adjacent windows would show similar read length fluctuations to G1 or G2 cells.

Could it be that FACS-sorted S phase cells that are in G1 according to in silico sorting are in fact very

early or very late S phase cells ? The data shown on Fig S1 do not convincingly eliminate this possibility, because read count profiles are shown for only one chromosome of only two cells.

First, the authors should show how the MAPD of early, mid and late S phase cells varies with window size (at constant coverage) compared to G1 cells.

Second, they should show an aggregate copy number profile of "computationally G1 cells in the S fraction", just as they did for the "computationally S cells in the G1 fraction" in Fig S1d, in order to evaluate the proportion of truly S phase cells present in the "computationally G1" population. By the way, please label the Y-axis in the replication profiles shown on Fig 1d and Fig S1d.

Third, they should show the distribution of the % of genome replicated, as estimated by the HMM, of the cell populations whose in silico sorting was discrepant with FACS.

Fourth, they may re-sort the FACS sorted-fractions in order to assess cross-contaminations directly by FACS.

Overall, the proposed experiments should more firmly convince the reader that their computational sorting is more reliable than FACS.

Minor criticisms

5. Cells computationally identified as G1 cells may in fact correspond to G2/M cells, especially those found in the late S FACS fraction. In fact, Fig S1a (compare G1 and G2 panels) suggest that it is practically impossible to computationally distinguish G1 from G2 cells using MAPD. Could the authors state and discuss this point where appropriate in the manuscript ?

6. Fig. 1a and Fig S1b,c show raw read counts per 200 kb windows of single cells, but the Y-axes are not labeled. What are the counts ? Are the counts in the inferred 4N regions really double of those in the inferred 2N regions ? Please label the Y axes and show the 0 positions.

7. The raw counts profiles shown in Fig 1a and Fig S1b,c, show a continuum of counts suggesting a lack of clear demarcation between the 2N and 4N regions. Please show the genome-wide distribution of raw read counts per 200 kb windows (or per uniform-coverage windows defined in G1 cells) to reveal if the distribution is biphasic, triphasic or continuous.

8. Legend to Fig 1d, what is the Y-axis scale for the green profile and the dark profile ? Were the profiles directly superimposable or did the authors perform any rescaling ? Please explain what the white vertical lines in the pile-up plots correspond to. The pile-up structures would be better named "triangular" than "conical". Please indicate on the Y-axis the % Replicated, as was done in Fig 2. In Fig 2a, the early replicating cells are more abundant than the late-replicating cells. Why ? Is it also the case for Fig 1d ? In contrast, in Fig 2b, a very different distribution of replicated fractions is observed. Why ?

9. Line 180 : " After filtering out cells that were not replicating or for which a two-fold relationship 181 was not observed between copy-number states, we analyzed 2,428 single GM12878 cells." Question : 2,428 out of how many (presumably asynchronous) cells ? What about the filtered cells ? Could they be assigned to G1 or G2 or was the MAPD ambiguous ? These informations are important to evaluate the fraction of computationally analysable cells in a growing population. As a control, can they show the same pileup plots from the eliminated cells ?

Line 195 : same question. 759 cells out of how many ? What is the profile of eliminated cells ?

10. Line 241, " we found that 49.6% of mappable genomic windows were called as a probable initiation site in at least one cell." Windows of which size ? How does this % change with window size?

11. Fig 3c,e,f, please explain in the legend what are the numbers in kb to the right of each panel. Are these the lengths or IRs as defined in the legend to Fig 3c ? Do the dotted lines in 3e and 3f represent the 25th and 75th percentiles of the range of track center positions ? I can clearly see one (3e) and several (3f) green tracks whose midpoint does not, contrary to what is stated in the legend, fall within the indicated 60kb (140 kb) range.

12. Figure 4b, their nomenclature "on time/yet to fire/premature/delayed" was not immediately clear to me. Am I correct that "on time" are regions that fired as expected, and "yet to fire" regions that did not fire and were indeed not expected to fire ? I would have preferred "fired as expected" and "unfired as expected" but if they decide to keep their nomenclature, please explain their meaning in the legend.

13. I agree that their data challenge the existence of large, constant replication timing regions (CTRs) in early replicating domains. This is consistent with previous cell population studies that mapped closely spaced but delimited initiation zones in regions previously described as early CTRs (Petryk, Tubbs, Wang, as cited above, please quote as appropriate). However, I don't think that their data exclude the existence of large CTRs in very late-replicating DNA. This is mainly because they have no or very few cells in the 80%-100% replicated range, when such late CTRs replicate. Presumably, cells in very late S phase resemble G2 cells and are missed by MAPD analysis. I suggest deleting the corresponding sentence lines 573-574.

15. Constitutively late IRs: what is the % of cells of different replicated fractions in which they fire ? Is the probability of firing independent of time already spent in S phase ?

16. Line 620 not clear if they take into account origin interference in their discussion of why inefficient IRs have a lower efficiency of initiation in late than in early S phase. I cannot find in the data where they show that "the majority (63.6%) of the inefficient IRs have a low probability of firing in early S phase, and a negligible probability of firing later in S phase".

Reviewer #3 (Remarks to the Author):

This report by Massey and Koren, utilizes single cell DNA sequencing to map DNA replication at high resolution. The authors reasoned that current approaches using the 10X or DLP platforms can routinely provide copy number information which can reveal replication origin use and S-phase dynamics in individual cells. Massey and Koren develop a creative approach (MAPD) to normalize the sequencing data and then extract genomic regions with increased copy number likely as a result of DNA replication. They argue – and convincingly demonstrate – that their method allows them to perform “in silico flow cytometry” which basically ranks cells according to DNA copy number. With this logic the authors are then able to test some basic, yet not well understood parameters of origin use in S-phase. They

demonstrate that most origins are fired from predictable sites in the genome with predictable timing. However, they find that some origins fire at seemingly random times through S phase suggesting that origin use is unlikely to be deterministic.

This report is very nicely put together: the data is of high quality and the conclusions appear to be robust. The methodology described offers an excellent framework that will stimulate future investigations into the control of origin use in S-phase.

A limitation of the work concerns the ambiguity of the resolution of the approach. It is unclear how large a replicon would need to be to be routinely detected and whether this limit applies evenly across the genome. As such it is difficult to know how many origins contribute to the IRs; or whether seemingly clustered initiation sites are actually part of the same replicon but appear separate due to detection issues. I note that the authors do discuss these limitations, but the manuscript may be improved if they could provide some statistical measures.

Minor Comments:

Figure 1D: the "1 fraction" label on the graph was confusing, I assume this is simply copy number? If so, it would benefit from a defined y-axis.

Figure 3C: I assume that each of the cells are represented by a single line, why are the lines in the 4-cell plot different thicknesses?

Figure 4H: Should the X-axis span from 1 to -2?

General comments:

We thank the reviewers for their kind words about our work and for their suggestions for improving the manuscript, which were very useful. In response to the reviewers' suggestions, we have added several additional analyses and supplementary figures to further clarify the data and support our claims. We also now analyze data from an additional cell line (COLO-829) to demonstrate the utility of *in silico* sorting and have made many text clarifications and improvements throughout the manuscript per the reviewers' suggestions.

While responding to the reviewers' questions about *in silico* sorting, we also identified and fixed an issue in our approach that impacted a subset of libraries. Specifically, we have changed the initialization strategy for the expectation-maximization procedure that identifies two linear models for G₁/G₂ vs. S-phase cells. Previously, we had initialized this with random assignment of cells to two populations. However, we discovered that a few of the libraries were sensitive to these initialization parameters and would give inconsistent results when the data were re-processed. To resolve this, we have switched to a deterministic initialization, creating initial assignments to the two populations based on scaled MAPD values. We have also changed the way that we determine which of the two linear models corresponds to G₁-phase cells: we now assign the model with a smaller predicted MAPD at the maximum coverage as "G₁", rather than at the minimum coverage. This change impacted six of the libraries: the G₁ and G₂ FACS fraction GM12878 libraries, one of the unsorted GM12878 libraries, one of the H1 libraries, and the H7 and H9 libraries. The effect of this update is reflected in the figure below for the GM12878 G₁ library:

Reviewer Figure 1. **Differences in cell cycle assignments for the GM12878 G₁ library after revision.** *In silico* sorting was performed with random initialization (v1) or MAPD-based initialization (v2). Each dot represents one cell, and linear models fit by expectation-maximization are shown (black lines). For a subset of samples, the linear model fit to the S-phase cells (steeper black line) was driven by a few extreme cells in v1. This has been corrected in v2.

As a result, we have updated the numbers presented throughout the manuscript and in the supplementary figures. All corrected values were extremely similar to the original values (*e.g.*, 50% → 49%), with the exception of the % cross-contamination of the GM12878 G₁-phase FACS fraction presented in Supplementary Fig. 1a. (We discuss this specific result in depth in a new paragraph on p. 5 of the revised manuscript.) None of the corrections influenced any other results or conclusions.

Per the journal's editorial formatting request, we have also shortened the title, abstract, and several of the section headings and figure legends.

Reviewer #1:

The authors utilized scWGS to study DNA replication profiling, that enables more accurate and more sensitive characterization of replication origin firing (initiation regions), compared to traditional flow cytometry. Using this new approach, the authors confirmed existing knowledge by demonstrating that replication initiation events in single cells are mostly the same as those observed in the population average. But they also identified a subset of initiation regions that are outliers that fire constitutively throughout the S phase. The new approach and the new scientific findings included in the manuscript are worthwhile for publication and will be of great interest to the relevant community. Before that, several aspects should be improved to make the manuscript more solid scientifically and better in overall quality.

1. In P4, opposite to the authors' claim, MDA is actually worse than DOP-PCR or DLP for CNV analysis. The key advantage of 10X Genomics platform is its high throughput, enabling the analysis of thousands of single cells simultaneously in one run.

We agree with the reviewer that the primary advantage of the 10x Genomics approach over previous methods is its throughput and have revised the text on p. 4 accordingly. We nonetheless find it useful to cite, as a secondary advantage, recent studies that have shown better performance of newer commercial MDA-based kits over DOP-PCR, in terms of both genome coverage and genome uniformity (*e.g.*, manuscript ref. 19: Gonzalez-Pena *et al.* 2021; DOI: 10.1073/pnas.2024176118).

2. In P6, I agree there are many benefits of using single-cell whole-genome sequencing to profile DNA replication, compared with traditional FACS. But it is still better to list the cons as well as the pros the authors advocated, such as a higher cost and a longer time for experiment procedure and data processing.

We thank the reviewer for the suggestion regarding this claim (which is now on p. 4). We have clarified that this text refers to cell cycle analysis using FACS vs. *in silico* sorting based on single-cell sequencing. We also now detail the pros and cons of single-cell replication timing to, *e.g.*, bulk analysis of FACS-sorted cells in the discussion (p. 14). This includes an acknowledgement of the additional cost, time, and analytical methods required.

3. In P7, the authors suggested that the symmetric track is due to bidirectional replication fork progression from a single origin. How to rule out the scenario of multiple origins fired in each observed track? The authors came to this point later by choosing <1Mb tracks, but here when the phenomenon is described in the manuscript for the first time, it is better to clarify a bit.

We agree with the reviewer that both scenarios (a single origin or multiple origins) are plausible and supported by the data, as we have stated later in the text. Per the reviewer's suggestion, we clarify this point at the first instance in which the tracks are described (p. 6).

The major concern is that, because single-cell WGS generate genome-wide noise and bias on top of the DNA replication profile, the authors have to be very careful in data interpretation, especially the analysis leading to the scientific conclusion that "there are two subsets of initiation regions" highlighted in the abstract. Along this line, I have several suggestions as following:

4. In P4, for the 2.4% of cells in G1 phase by FACS but in S phase by sequencing, and 25.1% of cells in S phase by FACS but in G1 phase by sequencing, the authors seem to be confident to claim that this discrepancy suggest single-cell DNA sequencing provides a better sensitivity. While this likely is true, to be more rigorous, it will be ideal to analyze just these cells with different assignments between FACS and sequencing, and make sure their replication profile and other properties are just the same as other cells with the same assignment from FACS and sequencing.

We have added a new paragraph on p. 5 to more clearly articulate the value of the *in silico* sorting assignments, and how they should be interpreted. Because *in silico* sorting is used here to define a control subpopulation, it is intentionally conservative in assigning cells as "G₁/G₂" (assigning all borderline cells as "S"). These assignments are then refined by copy-number inference using the HMM. For the GM12878 G₁ fraction, 323/1331 cells (24.3%) are called as "S" by *in silico* sorting, and 63 of these cells (4.7%) are confirmed S-phase contamination after the final label refining.

We previously illustrated this cross-contamination of replicating cells in the G₁ fraction (based on *in silico* assignments) in Supplementary Fig. 1d. We have added two additional panels to Supplementary Fig. 1 to further support this claim by specifically analyzing the 63 cells identified as replicating after HMM processing. We show that these cells are early S-phase cells (Supplementary Fig. 1e), consistent with contamination during FACS, and that their replication profiles are indistinguishable from early S phase cells isolated in the S-phase FACS fraction (Supplementary Fig. 1f).

We have also added an additional supplementary analysis of a published 10x Genomics dataset for the human melanoma cell line COLO-829 (manuscript ref. 23: Velazquez-Villarreal *et al.* 2020; DOI: 10.1038/s42003-020-1044-8). In this dataset, FACS was used to sequence only G₁-phase cells. We demonstrate that *in silico* sorting labels 233/1475 cells (15.8%) as "S" (Supplementary Fig. 4a), and that the S/G₁ aggregate profile based on *in silico* assignments is correlated with the bulk replication-timing profile for COLO-829 ($r = 0.79$; Supplementary Fig. 4b). Processing of these cells by the HMM confirmed that 32 cells (2.2%) were in S phase, and we show the corresponding single-cell data in Supplementary Fig. 4c.

Differences in experimental procedure (including the specific dye used for cell sorting) between the GM12878 and COLO-829 G₁ libraries may explain the difference in cross-contamination frequency between the two datasets (4.7% and 2.2%, respectively). However, we are cautious about drawing conclusions about the *frequency* of FACS cross-contamination from a handful of samples. Nonetheless, we believe these new analyses demonstrate convincingly that the methods we present can identify genuine instances of mis-sorting.

(Note: see the general comments above for a thorough explanation of why the *in silico* values for the GM12878 G₁ library have changed from 2.4% to 24.3%. This is the only major change in the results in this revision.)

I also wonder if we narrow the FACS gating in S phase (i.e., gating mid-S only), will that reduce the 25.1% number? 25% seems to be such a large number.

The full-S FACS fraction has a high degree of cross-contamination (25.1% based on *in silico* sorting; 31.8% after the HMM), likely because (as the reviewer suggests), it is contaminated on both sides of the gate, i.e., from both G₁ and G₂ cells. We agree with the reviewer that gating for mid-S only would likely reduce cross-contamination.

5. In P6, I wonder how robust the analysis result could be, when changing parameters in HMM fitting?

The only user-defined parameter in the HMM itself is the number of hidden states – given that we considered replication status to be binary, we have used a two-state model. For a particular pair of read counts and initial state assignments, the Baum-Welch algorithm was used to find maximum likelihood estimates for the transition and emission probabilities. (Cells for which the Baum-Welch algorithm did not converge on transition and emission probabilities after 500 iterations were excluded. We defined convergence with a tolerance 1×10^{-6} .)

6. In P9, is the arbitrary threshold of 1Mb sensitive to the analysis outcome? If change 1Mb to another number, will the conclusions still hold true?

The arbitrary threshold of 1Mb is used only to calculate a single value: that 49.7% of mappable 20kb windows are called as an initiation site in our “first approximation” approach. When we cluster tracks into IRs, we do not use the 1Mb threshold, precisely because it is arbitrary.

This particular result is sensitive to threshold: as we increase the threshold to include longer replication tracks, we call more windows as potential initiation sites. At the same time, increasing this threshold increases the likelihood that we are including tracks that represent the activity of multiple replication origins (adding noise into the analysis). The table below presents the result for different thresholds:

threshold	windows called as initiation sites	replication tracks included
300Kb	27.8%	11.2%
500Kb	39.2%	23.1%
1Mb	49.7%	46.5%
2Mb	56.9%	69.9%
No threshold	66.9%	100%

Importantly, the main conclusion – that replication initiation occurs at many sites across the genome – is robust across parameters. We chose 1Mb for this initial analysis because ensemble replication-timing profiles display strong peaks ~1Mb apart.

The same applies to later analysis when the authors cluster tracks into IRs by prioritizing shorter tracks over longer tracks. I wonder whether the analysis result is robust when changing such prioritizing parameters.

We suspect that the word “prioritize” was not sufficiently precise to convey how track length was used in the IR-defining algorithm. We have rephrased the explanation of the algorithm on p. 7 to remove this word and to explain the procedure more clearly. We reproduce that explanation here:

“Replication tracks that overlapped multiple shorter replication tracks were treated as agnostic to IR location because they could plausibly be explained by firing of multiple of the overlapped origins or of a single central origin. These uninformative tracks were thus excluded from use in clustering.”

7. For DLP+ data, it is good to see it can reproduce the replication timing profile. But will it generate the same conclusions as the authors described later in the paper? Especially the surprising conclusion highlighted in the abstract, that there is a subset of “ensemble late” initiation regions that can fire randomly throughout the S phase.

Yes, although noisier, the DLP+ data for GM18507 do generate the same conclusions as the 10x Genomics data for the 9 cell lines, including the subset of randomly firing “ensemble late” IRs. We have added the DLP+ results for all applicable analyses into Supplementary Fig. 10 and Supplementary Fig. 11.

Reviewer #2:

Recent studies have demonstrated the power of single cell DNA sequencing for determining their DNA replication program based on the doubled copy number of replicated compared to unreplicated DNA sequences. However, these early studies were limited in resolution and cell number.

Here, Massey and Koren report sequence data from thousands of single cells. They describe an *in silico* strategy to discriminate S phase cells from other cell cycle stages, which they suggest is more accurate than traditional flow cytometry, and simplifies experimental procedures. They find that single cells initiate replication at consistent initiation regions (IR) < 100kb. In contrast, the firing time of IR is largely variable from cell to cell. IR that on aggregate fire early in S phase tend to fire early in S phase in all cells, but IR that on aggregate fire late in S phase can with some probability fire at any stage in S phase. Only 10% of late IR appear to fire constitutively late. Finally, by analyzing different cell types with principal component analysis, they show that single cells follow cell-type specific trajectories of S-phase progression.

Overall this is a technically strong and well-written story that advances the state-of-the-art of single-cell replication timing analysis. However, the gain in biological knowledge is at best marginal. Several studies, both at the cell population and single molecule or single cell level, have previously reported the locations and replication time distributions of initiation regions in human cells and have proposed based on these data that replication time largely emerges from the stochastic firing of origins with different firing probabilities. While confirmation at the single-cell level is nevertheless important, no effort has been presented here to evaluate the consistency of e.g. IR location with previous IR mapping studies. Furthermore, some of their conclusions seem to be incorrect, as explained below, and several Figures are insufficiently legended.

We thank the reviewer for the detailed critique of our work and many useful suggestions. As the reviewer notes, a number of methods have been previously reported for identifying the locations of replication origins from populations of cells and from single molecules. We have sought to clarify throughout the text that we are not presenting an origin-mapping method and that we are not claiming to have identified a comprehensive list of replication origins in human cells. Furthermore, comparisons of these origin-mapping methods to ensemble replication timing have already been published – given the concordance of our initiation regions with ensemble replication timing profiles, we feel that direct comparison of IRs to existing lists of origins would be redundant and not provide new biological insight.

In this manuscript, we present the most comprehensive empirical test of replication timing consistency across cells to date. In doing so, we demonstrate the biological value added of higher throughput experiments in uncovering rarer variation. While our results are consistent with the general concept that replication timing largely emerges from stochastic firing of origins with different firing probabilities, we find evidence that the proposed models for this are missing key details about the kinds of variation that exist among cells. Namely, stochastic origin firing, as it has been previously described, does not explain the existence of multiple subtypes of “ensemble late” replication origins, which we report here for the first time.

Major criticisms:

1. The studied cell lines are diploid. Unless origin firing is strictly coordinated between the two homologous chromosomes of a pair, one expects at any time in S phase a significant fraction of genomic segments to be replicated on one homolog but not on the other one (copy number = 3N). However, replication profiles were inferred for each cell using a Hidden Markov Model that only considers two possible states: both alleles are unreplicated (2N) or both are replicated (4N). Why did not they run a three-state HMM rather than a two-state HMM?

We thank the reviewer for this excellent question, which we feel many readers may share. It is true that stochasticity in replication initiation makes it likely that homologous origins will fire out of sync, producing regions of 3N. However, we found that the per-cell coverage in our dataset was not sufficient to reliably distinguish the up-to-1.5x difference in read counts between 2N/3N or 3N/4N. Thus, a three-state HMM appeared to provide suboptimal results. Furthermore, prior analyses of asynchrony in single-cell replication indicate that it is relatively uncommon and that the time between replication of homologues is short (Dileep and Gilbert, 2018; Takahashi *et al.*, 2019). Both previous analyses were performed using allele-specific mapping in hybrid mice. Thus, the size of asynchronous regions and temporal difference between alleles make it challenging to detect 3N regions with our current approach, and in human cells, where SNP density is lower. Future work with higher per-cell coverage and/or allele-specific mapping will be better powered to detect three copy-number states. In a new paragraph in the discussion (p. 14), we elaborate on limitations of our approach, including this one.

Furthermore, in the Methods paragraph "Replication state inference", they mention that "If this initial model did not converge, or if the ratio between the two mean copy numbers was not ~2 (between 1.5 and 2.5), the cell was excluded" and that "we interpolated the data back onto fixed size 20kb windows. Windows for which a value other than 2 or 4 was interpolated were masked". Could it be that these filters specifically eliminate cells and/or regions with significant replication asynchrony between homologues?

The first metric excludes entire cells based on the expected copy-number ratio between 2N and 4N regions genome-wide. Thus, it is an assessment of the quality of the HMM output, using orthogonal information. We inferred copy number using a two-state HMM and downstream analyses assumed that these two states corresponded to 2N and 4N. Thus, although the reviewer is, in principle, correct that a cell with significant asynchrony would be excluded in this step, its removal would be appropriate – the two states detected by the HMM in this hypothetical cell would not correspond to 2N and 4N and would thus not be appropriate for comparison to the other cells. We have added a sentence to the methods section (p. 17) expanding on the variety of alternative explanations for cells that fail this quality-control step.

The second metric, which filters individual windows in individual cells, is unrelated to copy-number inference. Instead, it handles the situation where a fixed size window overlaps

multiple G_1 -defined windows. In cases where the overlapped G_1 windows do not share a copy-number state, the fixed size window will have a non-integer copy-number because we lack the resolution to define the precise location of the replication fork. We have clarified this in the methods section (p. 18).

2. More detailed comparison of their initiation regions (IRs) with those seen by other techniques (OK-seq, Petryk et al 2016; EdUseq-HU, Tubbs et al 2018; high-resolution Repli-seq, Zhao et al 2020; Optical Replication Mapping, Wang et al 2021) is required. In particular, since they identified 7482 IRs in GM12878 cells, which are very closely related to GM06990 cells used in OK-seq experiments, a direct comparison is feasible. It is not clear whether their analysis brings any novel information about initiation regions.

While we agree that these comparisons are interesting to make, we think that they are redundant with analyses already reported in the literature and likely will not yield any new insights. Our single-cell IRs strongly overlap with ensemble replication timing peaks, which have themselves already been compared to the cited papers. Thus, our single-cell IRs will correspond well with origins/peaks described by these methods. We have indeed confirmed this visually and see little added value in reporting a formal analysis.

We also wish to clarify that we do not present single-cell replication timing profiling as a method to map novel replication origins or to identify their locations at greater precision than other techniques. More generally, we do not claim to map replication origins here at all. Indeed, we believe that single-cell replication timing profiling is noisier for this purpose than some other approaches. Rather, its advantage is in accurately describing the different forms of heterogeneity in replication initiation.

We have added a few sentences to the discussion (p. 15) acknowledging that these other methods are more appropriate for origin mapping. We reproduce that here:

Single-cell sequencing is, at least currently, not the optimal technology for identifying individual replication origins, and existing origin-mapping methods (*e.g.*, OK-seq [30], HU-Edu-seq [35], high-resolution Repli-seq [10], or optical replication mapping [7]) are better suited to this purpose. Origins identified by these methods overlap well with ensemble replication-timing profiles, as do the IRs we identified here.

3. The identified number of initiation events is problematic. It is generally accepted that some 20,000 - 50,000 initiation events take place during each replication cycle in human cells. However, they claim (line 320) that " 2,640 IRs (35.3%) accounted for 90% of the replication tracks, indicating that about a third of the IRs are responsible for the vast majority of initiation events genome wide". If so, only 3,000 initiation events would take place per S phase per haploid genome, and the average spacing between initiation events would be about 1 Mb, which is one order of magnitude larger than repeatedly observed by DNA fiber techniques. In fact, given a fork speed of 2 kb/min, if initiation events were regularly spaced every Mb, it would take a single fork 8h to travel and converge with a fork emitted from a neighboring origin. These kinetics are inconsistent with many studies over the past decades.

Clearly, the authors are underestimating the number of initiation events per S phase by a factor of 10. Why?

The discrepancy can be attributed to several factors, mostly technical:

First, IRs may represent multiple origins: within a single cell, a replication track may reflect multiple fired origins; across cells, the width of an IR may reflect heterogeneity in origin position. Thus, our method underestimates the number of initiation events proper, while some previous methods count many clustered initiation events as separate.

Second, as explained above, we agree that we have not identified a comprehensive list of all replication origins and wish to clarify that we do not see single-cell replication timing as the best method for doing so. We have added two sentences about this to the discussion (p. 15), which are reproduced above. In addition, we have rewritten the sentence the reviewer quoted above (p. 8), which we realize did not convey our intended meaning clearly:

“2,595 IRs (34.5%) accounted for 90% of all replication tracks, indicating that about a third of the IRs are used consistently across cells.”

Third, our data are consistent with ensemble replication timing data, which also reveal prominent peaks ~1Mb apart, while not ruling out other, more rarely used origins.

Finally, it is true that a replication fork moving ~2Kb/min would take ~8h to travel 1Mb. Therefore, two replication forks that originate 1Mb apart and move at that speed would be expected to converge halfway between the two origins in ~4h. This is consistent with the ~8 hour S-phase observed in most human cell lines.

Can they estimate the apparent speed of replication progression from their pile-up plots and compare with known fork speeds?

We would be very cautious about estimating the speed of replication progression from single-cell replication timing data, for several reasons. First, it is unknown how to translate the percentage of the genome replicated into clock time: doing so requires assuming we know precisely how long S-phase lasts, that this period is the same in every cell, and that each proportion of the genome replicated corresponds to a uniform duration of time. Second, we can only track replication fork progression until neighboring replication forks converge, which means we often have limited information about individual loci. Third, this approach to estimating replication fork progression considers replication tracks in different cells to be different time points along the same trajectory. In other words, it assumes that every cell fires the same origins in the same order. In contrast, our data show that overall S-phase progression in a cell does not fully predict replication progression at a particular locus. Thus, cells later in S-phase sometimes display *less* replication progression at a locus than cells earlier in S-phase. This underscores that, although replication track length

corresponds to the amount of time a replication fork has been active within a cell, we only have data for one time point per cell.

They suggest (line 515) that "increasing the number of cells analyzed will likely yield additional IRs in all cell lines and also further narrow their localization". If so, it is likely that they only detected a fraction of all initiation events in the 2,428 GM12878 cells. Could they subsample this and detect origins at increasing numbers of cells? If origin number does not saturate at 2,400 cells, origin description cannot yet be comprehensive. If on the other hand origin number is approaching a plateau of 3,000 initiation events, then resolution of the sequencing data is probably the limiting factor in origin detection and increasing the coverage is required before a definitive description can be proposed.

We thank the reviewer for this helpful suggestion for more directly demonstrating the effect of sample size on the number of IRs identified. We have performed the suggested analysis (Supplementary Fig. 9), which supports the claim that we observed fewer IRs in the additional cell lines relative to GM12878 because of the relatively smaller sample size. This suggests we have not captured all initiation events due to the number of cells, confirming the reviewer's suspicion that part of the discrepancy between the number of IRs we identified and the number of origins previously reported is the sample size.

4. I was not entirely convinced that their *in silico* sorting method performs better than FACS for several reasons.

We have added a new paragraph on p. 5 elaborating on the cross-contamination detected by *in silico* sorting in our GM12878 FACS fractions, as well as two new analyses of these cells (Supplementary Fig. 1e-f). In addition, we have added data from a separate published FACS experiment, where cross-contamination is again observed (Supplementary Fig. 4). See below, and response to Reviewer 1 Comment 4, for more detail.

a) To computationally distinguish replicating cells from non-replicating cells, they quantify local read depth fluctuations using a metric called MAPD (median absolute deviation of pairwise differences between adjacent genomic windows), of which they say nothing for the unfamiliar reader. Please give a reference. In particular, it remained unclear to me why "scaling" (do they mean multiplying or dividing?) the MAPD by the square root of read depth produced a linear relationship between coverage and scaled MAPD. I would have expected that the scaled MAPD would no longer vary with coverage, at least for G1 cells.

We thank the reviewer for pointing out the missing reference for MAPD, which has been added to the first mention of this metric in both the results and method sections. We have also added two additional clarifications: (1) the x-axis of Figure 1b displays the average number of reads per Mb (calculated per-cell, not per-window), and (2) MAPD is "scaled" by dividing by the square root of reads/Mb. This is done to linearize MAPD relative to reads/Mb.

As the reviewer anticipates, scaled MAPD indeed does not vary much with reads/Mb in G₁ cells (slope close to 0). However, per-window read counts in G₁ cells with high coverage still fluctuate less than those with low coverage, because read counts are sampled from a Poisson distribution. The linear model fit to G₁ cells allows us to calculate the expected MAPD value for a given reads/Mb value and detect cells that deviate strongly from that expectation.

b) The choice of window size is critical for using the MAPD to distinguish between replicating and non-replicating cells. The chosen window size was 1Mb. The discriminatory power should therefore be maximum when replicated and/or unreplicated segments alternate at a characteristic length of 1Mb. This is probably the case in mid S phase. At the beginning or end of S phase, however, the characteristic lengths of replicated and unreplicated segments may be very different, so that most pairs of adjacent windows would show similar read length fluctuations to G₁ or G₂ cells.

We have clarified in the text that we used *in silico* sorting (*i.e.*, MAPD) only to identify cells to use as G₁/G₂ controls for defining windows. After counting reads in G₁-windows for each cell, replication states were inferred by the HMM for all cells (including those *in silico* assigned as G₁/G₂). Thus, a cell very early or very late in S-phase that has been incorrectly assigned as G₁/G₂ based on MAPD will have no impact on downstream analyses unless it changes the window definitions. In the figure below, we show that *fewer* cells are assigned as G₁/G₂ as we increase window size, across all FACS fractions.

Reviewer Figure 2. **Fewer cells are called as G₁/G₂ as MAPD window size increases.** *In silico* sorting was performed for each sorted GM12878 library, using MAPD values calculated for 200Kb, 500Kb, or 1Mb windows. Bars represent the proportion of cells in the library called as G₁/G₂ under each parameter. The 1Mb windows were used in the manuscript.

Could it be that FACS-sorted S phase cells that are in G₁ according to *in silico* sorting are in fact very early or very late S phase cells? The data shown on Fig S1 do not convincingly eliminate this possibility, because read count profiles are shown for only one chromosome of only two cells.

Yes, it is likely that we are losing cells at the very beginning and very end of S phase. However, this is not due to the *in silico* sorting – as explained above, after *in silico* sorting,

these cells will still be considered as potentially replicating and processed by the replication-state HMM. (We have clarified this point on p. 5.)

The reason that the earliest and latest S-phase cells will be lost is that replication, copy-number aberrations, and technical noise each impact the observed read depth. It is only in the context of consistent copy number changes at multiple loci across the genome that we are confident that the fluctuations we observe reflect replication.

The new Supplementary Fig. 2 includes the distribution of read counts for 40 example cells in the S-phase FACS fraction, and whole-genome plots for five of these cells (analogous to those shown in Supplementary Fig. 1b, but for the whole genome). This includes distribution plots for 8 example cells called as G₁/G₂ (all unimodal) and one whole-genome plot (uniform read depth). We further direct the reviewer to Reviewer Fig. 3 below.

First, the authors should show how the MAPD of early, mid and late S phase cells varies with window size (at constant coverage) compared to G₁ cells.

As shown in the bar graph above, smaller window sizes than the one used result in *more* cells being called as G₁/G₂. This is true for all five FACS fractions, including the early S- and late S-phase cells. We believe that this should alleviate the concern that the window size used was too big to capture replication occurring in early and late S phase.

Second, they should show an aggregate copy number profile of "computationally G₁ cells in the S fraction", just as they did for the "computationally S cells in the G₁ fraction" in Fig S1d, in order to evaluate the proportion of truly S phase cells present in the "computationally G₁" population.

We have clarified in the main text and in the methods that *in silico* sorting assigns only those cells that are unambiguously not replicating as "G₁/G₂", because the goal is to identify a high-confidence subset of cells to use as a control for sequencing biases and cell-line-specific copy-number variants. Indeed, computationally assigned G₁/G₂-phase cells were predominantly (if not exclusively) non-replicating, as observed as uniform read depth across the genome. This is shown in the top panel of Reviewer Figure 3.

Reviewer Figure 3. **Contamination of non-replicating cells in the S-phase FACS fraction.** Top panel: read counts in 20kb windows were summed across all cells labeled as “G₁/G₂” by *in silico* sorting within the S-phase FACS fraction. Read depth across the chromosome is uniform, consistent with the absence of DNA replication. Bottom panel: read counts summed across all cells labeled as “S” by *in silico* sorting within the S-phase FACS fraction are shown for comparison. Fluctuations in read depth are consistent with replication timing.

By the way, please label the Y-axis in the replication profiles shown on Fig 1d and Fig S1d.

We thank the reviewer for the feedback on the difficulty in interpreting the Figure 1d and Supplementary Figure 1d Y-axes. We have updated the Figure 1d axis to more clearly show that the top panel shows replication timing, whereas the middle and bottom panels show multi-cell fractions. We have also changed the Supplementary Figure 1d Y-axis label to “RT” to be consistent with other replication timing profiles throughout the manuscript.

Third, they should show the distribution of the % of genome replicated, as estimated by the HMM, of the cell populations whose *in silico* sorting was discrepant with FACS.

Cells inferred to be replicating within the G₁ FACS fraction are in early S-phase (<30% replicated). We now show this in Supplementary Figure 1e.

In addition, we have added the single-cell replication profiles from these cells, compared to cells <30% replicated in the S-phase FACS fraction (Supplementary Figure 1f).

Fourth, they may re-sort the FACS sorted-fractions in order to assess cross-contaminations directly by FACS.

Unfortunately, all of the sorted cells were harvested for DNA extraction and sequencing, so we are not able to repeat the sorting by FACS on the same cells.

Overall, the proposed experiments should more firmly convince the reader that their computational sorting is more reliable than FACS.

Minor criticisms:

5. Cells computationally identified as G1 cells may in fact correspond to G2/M cells, especially those found in the late S FACS fraction. In fact, Fig S1a (compare G1 and G2 panels) suggest that it is practically impossible to computationally distinguish G1 from G2 cells using MAPD. Could the authors state and discuss this point where appropriate in the manuscript?

The reviewer is correct that we cannot distinguish G₁ cells from G₂ cells. We have added a sentence to this effect on p. 5, and have changed references to “G₁ cells” to “G₁/G₂ cells” when describing *in silico* sorting. (All remaining mentions of “G₁ cells” refer to the FACS fraction.)

6. Fig. 1a and Fig S1b,c show raw read counts per 200 kb windows of single cells, but the Y-axes are not labeled. What are the counts? Are the counts in the inferred 4N regions really double of those in the inferred 2N regions? Please label the Y axes and show the 0 positions.

We have updated the relevant panels of Figure 1 and Supplementary Figure 1 to show the raw read count values on the y-axis. Across cells, counts in inferred 4N regions are not quite double those in inferred 2N regions (closer to ~1.75x). As described in the methods, we used the expected two-fold relationship between read counts in 2N and 4N regions to exclude cells that failed to demonstrate this relationship.

7. The raw counts profiles shown in Fig 1a and Fig S1b,c, show a continuum of counts suggesting a lack of clear demarcation between the 2N and 4N regions. Please show the genome-wide distribution of raw read counts per 200 kb windows (or per uniform-coverage windows defined in G1 cells) to reveal if the distribution is biphasic, triphasic or continuous.

The distribution of read counts differs cell-by-cell and the bimodal distribution is most visually apparent in mid-S phase cells, where there is a large proportion of replicated windows *and* of unreplicated windows. To address this comment, we have added Supplementary Fig. 2, which displays the read count distribution for 40 randomly selected single cells, all from the S-phase FACS fraction library. This includes 8 cells that are not replicating, and 32 cells spread across S phase.

8. Legend to Fig 1d, what is the Y-axis scale for the green profile and the dark profile? Were the profiles directly superimposable or did the authors perform any rescaling?

The two lines in the top panel of Figure 1d represent replication-timing profiles, which are directly superimposable. We have changed the y-axis label to reflect this, and also added in y-tick values to indicate the scale. We hope these changes clear up this confusion.

Please explain what the white vertical lines in the pile-up plots correspond to.

The white vertical lines correspond to low-mappability regions that were masked, as described in the methods. We have added a sentence to the legend for Figure 1d to explain this.

The pile-up structures would be better named "triangular" than "conical".

Our intention was to emphasize the vertical symmetry of the pileups, but agree that "conical" rightly refers only to three-dimensional objects. We have changed the text to refer to these structures as "triangular".

Please indicate on the Y-axis the % Replicated, as was done in Fig 2.

We thank the reviewer for pointing out that this panel was not labeled clearly enough. We added the appropriate y-axis label ("Fraction") and added y-tick values to indicate the scale.

In Fig 2a, the early replicating cells are more abundant than the late-replicating cells. Why? Is it also the case for Fig 1d? In contrast, in Fig 2b, a very different distribution of replicated fractions is observed. Why?

The distribution of cells across S-phase in Figure 2 is skewed by FACS.

Figure 2a includes GM12878 cells from 8 populations (G_1 , G_2 , early S, late S, full S, and three unsorted populations). As shown in Supplementary Figure 1a, the "late S" fraction contains a large number of G_2 cells, while the majority of cells in the "early S" fraction are in S-phase. This skews the population toward early S-phase cells.

In contrast, Figure 2b includes GM18507 cells from four populations (G_1 , G_2 , S, unsorted). Differences in FACS gating in this previously published dataset resulted in the enrichment of mid-S phase cells observed in Figure 2b. (Unfortunately, we do not know which cell came from which FACS fraction in this dataset.)

In Figure 1d, each *in silico* "fraction" contains the same number of cells.

9. Line 180: " After filtering out cells that were not replicating or for which a two-fold relationship was not observed between copy-number states, we analyzed 2,428 single GM12878 cells."

Question: 2,428 out of how many (presumably asynchronous) cells? What about the filtered cells? Could they be assigned to G_1 or G_2 or was the MAPD ambiguous? This information is important to evaluate the fraction of computationally analyzable cells in a growing population. As a control, can they show the same pileup plots from the eliminated cells?

We analyzed a total of 9,580 GM12878 cells (detailed on p. 4 as 5,793 sorted + 3,787 unsorted). As mentioned in the text, 2,437 (25.4%) of these cells were analyzed for replication timing.

Of the remaining cells:

- 5,379 (56.1%) were assigned as “G₁/G₂” by the HMM
- 1,583 (16.5%) were removed because the HMM failed to converge (typically high sequencing noise or large technical artifacts; see (a) below)
- 181 cells (1.9%) were removed by quality-control filtering (primarily cells that are likely very late in S-phase; see (b) below)

Reviewer Figure 4. Cells filtered by quality-control steps do not display alternative replication profiles.

a Whole-genome read depth profiles for three GM12878 cells for which the HMM failed to converge. In each cell, there are multiple chromosomal and sub-chromosomal copy number abnormalities. Because it is unclear which copy number corresponds to the diploid state, abnormal regions cannot be filtered out. In total, 1,583 (16.5%) GM12878 cells were filtered at this stage. **b** Copy-number inferences for 181 (1.9%) GM12878 cells that failed quality control filtering. Many of these cells had unusual profiles at the extreme GC-content regions (e.g., Chr1:1-50Mb) or were fully replicated on some chromosomes but not others.

Across all unsorted libraries, we found that on average 21.6% of cells (range: 12.1-31%) were replicating and analyzable. (One GM12878 library failed to amplify properly and yielded only a handful of useable cells – this is excluded from the calculation of mean and range). We have added a sentence on p. 5 about the expected proportion of S-phase cells in an unsorted culture.

Line 195: same question. 759 cells out of how many? What is the profile of eliminated cells?

There were 3,040 cells in the DLP+ dataset, of which 759 (25.0%) were replicating and analyzable.

The breakdown of remaining cells was similar:

- 1,189 (39.1%) were assigned as "G₁/G₂" by the HMM
- 1,043 (34.3%) were removed because the HMM failed to converge
- 49 cells (1.6%) were removed by quality-control filtering

10. Line 241, "we found that 49.6% of mappable genomic windows were called as a probable initiation site in at least one cell." Windows of which size? How does this % change with window size?

We have clarified in the text (p. 7) that we are talking about 20kb windows, which is the highest resolution at which we confidently call replication states and the resolution at which we define IR boundaries. This value is specific to replication tracks less than 1Mb, but the broader claim that a large proportion of genomic windows holds true at different track length thresholds (see response to Reviewer 1, comment 6).

11. Fig 3c,e,f, please explain in the legend what are the numbers in kb to the right of each panel. Are these the lengths of IRs as defined in the legend to Fig 3c? Do the dotted lines in 3e and 3f represent the 25th and 75th percentiles of the range of track center positions? I can clearly see one (3e) and several (3f) green tracks whose midpoint does not, contrary to what is stated in the legend, fall within the indicated 60kb (140 kb) range.

Yes, the numbers to the right of each panel in Figure 3c, 3e, and 3f are the IRs widths, as defined in the legend of Figure 3c. We have added an explanation to the legend of Figure 3c.

Yes, the dotted lines in Figure 3e and 3f are the 25th and 75th percentiles of the midpoints, and not the extremes. Thus, some tracks have midpoints outside of the indicated ranges. We have revised the legend of Figure 3e to state that explicitly.

12. Figure 4b, their nomenclature "on time/yet to fire/premature/delayed" was not immediately clear to me. Am I correct that "on time" are regions that fired as expected, and "yet to fire" regions that did not fire and were indeed not expected to fire? I would have preferred "fired as expected" and "unfired as expected" but if they decide to keep their nomenclature, please explain their meaning in the legend.

We agree that the nomenclature we have used in Figure 4 is not ideal and thank the reviewer for the suggested change. We have changed the labels in the key for Figure 4b/c.

13. I agree that their data challenge the existence of large, constant replication timing regions (CTRs) in early replicating domains. This is consistent with previous cell population studies that mapped closely spaced but delimited initiation zones in regions previously described as early CTRs (Petryk, Tubbs, Wang, as cited above, please quote as appropriate). However, I don't think that their data exclude the existence of large CTRs in very late-replicating DNA.

This is mainly because they have no or very few cells in the 80%-100% replicated range, when such late CTRs replicate. Presumably, cells in very late S phase resemble G2 cells and are missed by MAPD analysis. I suggest deleting the corresponding sentence lines 573-574.

We thank the reviewer for this recommendation. We have changed the referenced to sentence (p. 13) to read:

“Analogously, our data do not support the existence of large constant replication regions (CTRs) [34], particularly in early-replicating regions, for which we have more data.”

In the sentence about initiation regions being on the scale of kilobases (p. 13), we have added citations of Petryk *et al.* 2016 and Tubbs *et al.* 2018 (in addition to the previous citations of Zhao *et al.* 2020 and Wang *et al.* 2021.

We also note that the IRs we defined are not skewed toward early-replicating regions relative to the entire genome, as seen in the figure below. Although we captured fewer late S-phase cells than early S-phase cells, IRs observed in late-replicating regions were also small, discrete segments.

Reviewer Figure 5. **Initiation regions are not strongly skewed toward early-replicating regions of the genome.** The distribution of replication timing values for the whole genome (blue) is compared to that of initiation regions (pink).

14. Constitutively late IRs: what is the % of cells of different replicated fractions in which they fire? Is the probability of firing independent of time already spent in S phase?

We defined constitutively late IRs as those that we never observed to fire in a cell less than 50% replicated. Thus, by definition, the probability of firing cannot truly be independent from time in S phase – within our observations, these IRs have a probability of firing of zero in cells 0-50% replicated. (We emphasize that this is an operational definition for the purpose of analysis, and not a claim about the underlying biology.)

For any region, the probability of being replicated increases with time already spent in S phase because the whole genome must somehow be replicated by the end of S phase. On the other hand, the probability of an IR firing (rather than being passively replicated) does not appear to compound with time spent in S phase.

Rather, for each constitutively late IR there was a particular decile of S-phase in which the probability of firing was maximized, suggesting that each of these IRs has a preferred firing time within the later half of S-phase. This is illustrated in the figure below. For this analysis, the probability of firing was calculated as the number of replication tracks supporting IR firing divided by the number of informative cells, within each decile of S-phase. (We considered a cell that had replicated the region containing the IR – but for which we had not called the IR as “fired” – to be uninformative.)

Reviewer Figure 6. **Constitutively late initiation regions (IRs) appear to have preferred firing times within late S-phase.** Cells were divided up into deciles of S-phase based on the percent of the genome replicated (<10% replicated, 10-19% replicated, etc.). For each constitutively late IR, the proportion of cells in each decile that had fired the IR was calculated. Most IRs have their maximum firing prior to the final decile of S phase, suggesting that their probability of firing does not compound over time.

- Line 620 not clear if they take into account origin interference in their discussion of why inefficient IRs have a lower efficiency of initiation in late than in early S phase. I cannot find in the data where they show that "the majority (63.6%) of the inefficient IRs have a low probability of firing in early S phase, and a negligible probability of firing later in S phase".

The data referenced is shown in Figure 5e: this sentence refers to the “early + rare” class of late IRs (27% of all IRs and 63.6% of late IRs). We have added a reference to the figure at the appropriate place in the text.

To the reviewer's broader question: passive replication across the region containing an "early + rare" IR cannot explain the decline in firing probability that we describe because these events occur in "ensemble late" regions. While these IRs are frequently passively replicated in late S phase, the previously unappreciated phenomenon we describe is that "ensemble late regions", which remain unreplicated in most early/mid S phase cells (actively *or* passively), contain IRs that sometimes fire early or not at all. In the cells that do not fire these IRs in early S phase, the region tends to remain unreplicated until late S phase.

Reviewer #3:

This report by Massey and Koren, utilizes single cell DNA sequencing to map DNA replication at high resolution. The authors reasoned that current approaches using the 10X or DLP platforms can routinely provide copy number information which can reveal replication origin use and S-phase dynamics in individual cells. Massey and Koren develop a creative approach (MAPD) to normalize the sequencing data and then extract genomic regions with increased copy number likely as a result of DNA replication. They argue – and convincingly demonstrate – that their method allows them to perform “in silico flow cytometry” which basically ranks cells according to DNA copy number. With this logic the authors are then able to test some basic, yet not well-understood parameters of origin use in S-phase. They demonstrate that most origins are fired from predictable sites in the genome with predictable timing. However, they find that some origins fire at seemingly random times through S phase suggesting that origin use is unlikely to be deterministic.

This report is very nicely put together: the data is of high quality and the conclusions appear to be robust. The methodology described offers an excellent framework that will stimulate future investigations into the control of origin use in S-phase.

1. A limitation of the work concerns the ambiguity of the resolution of the approach. It is unclear how large a replicon would need to be to be routinely detected and whether this limit applies evenly across the genome. As such it is difficult to know how many origins contribute to the IRs; or whether seemingly clustered initiation sites are actually part of the same replicon but appear separate due to detection issues. I note that the authors do discuss these limitations, but the manuscript may be improved if they could provide some statistical measures.

We thank the reviewer for this suggestion. We have added a new Supplementary Figure 3, in which we simulate realistic sequencing data from a defined “true” profile and then assess how accurately that profile is recovered by the HMM. While this simulation is somewhat simplified and does not attempt to capture the full complexity of our real data, we nonetheless find that the HMM performs well at recovering the “true” profile. Specifically, we find that errors in inferred replication state primarily affect the lowest-coverage cells and cells >99.5% replicated. We further demonstrate that inference errors are relatively uniform relative to ensemble replication timing. Looking specifically at the question of how large a replicon would need to be to be reliably detected, we find that we can detect >90% of replicons comprised of at least two genomic windows. Replicons comprised of a single genomic window are detected 73% of the time.

Minor Comments:

2. Figure 1D: the “1 fraction” label on the graph was confusing, I assume this is simply copy number? If so, it would benefit from a defined y-axis.

Yes, the y-axis for the top panel of Figure 1d is “replication timing”. We have stated that explicitly in the y-axis label now and added appropriate y-ticks. To help further clarify this confusion, we have also added the y-ticks on the middle and bottom panels of Figure 1d, where each row represents an aggregated fraction. The “titles” for these three panels have been moved to the right-sided y-axis.

3. Figure 3C: I assume that each of the cells are represented by a single line, why are the lines in the 4-cell plot different thicknesses?

We appreciate the reviewer pointing out the y-axis is cut off on both the top and bottom of this panel. This axis has been corrected.

4. Figure 4H: Should the X-axis span from 1 to -2?

We thank the reviewer for spotting this error. The x-axis of Figures 4h should match the x-axis of Figure 4c, and this has been corrected in our resubmission.

REVIEWERS' COMMENTS

Reviewer #1 (Remarks to the Author):

The authors have addressed most of my concerns and suggestions, however, there are still one major concern and one minor issue that is worth to mention for the quality of the paper.

The main concern is to follow my point 6 in the previous round, “when the authors cluster tracks into IRs by prioritizing shorter tracks over longer tracks, I wonder whether the analysis result is robust when changing such prioritizing parameters”. The authors addressed this concern by explaining in detail what “prioritize” means in their analysis pipeline, which is actually to remove longer tracks that overlapped multiple shorter tracks, indicating these are multiple firing origins, thus should not be considered. My concern is that, even after this filtering process, a certain portion of the remaining tracks could still be from multiple firing origins. Although no shorter tracks overlapping with them are observed, it does not guarantee such shorter tracks do not exist. The authors also acknowledged this possibility when addressing my point 3 in their rebuttal.

My suggestion here is that although there is no way to exactly separate tracks with single origin from those with multiple origins, at least the authors may try different filtering thresholds based on track length, in addition to the current filtering to remove tracks that are for sure from multiple origins. The goal is to make sure the most important scientific conclusions from this paper are robust and not affected by such parameter tuning in the analysis pipeline.

A minor issue is to follow my point 1 in the previous round. The PNAS paper the authors cited focused on PTA method, which is different from MDA. Although in the paper Figure 2 also compared between DOP-PCR and different versions of MDA assays, the comparison was all based on 1bp bin size. When comparing uniformity, noise, or the accuracy to detect CNV, the bin size could change the result drastically. For example, in the LIANTI paper (Science 356, 189-194, 2017), whose data the PNAS paper used a lot, Figure 1E showed CV of different methods, and suggested MDA is better than DOP-PCR in terms of uniformity/noise when bin size is <10Kb (same as PNAS paper Fig 2B), but worse than DOP-PCR when bin size is >10 Kb. That is, MDA could be better or worse than DOP-PCR, depending on later analysis of CNVs <10Kb or >10Kb. Usually in single-cell RT studies, most CNVs (replication tracks) will be >10Kb.

With that said, I do understand the authors need to claim MDA is better than DOP-PCR, since DOP-PCR has already been used to study single-cell RT before. Also, MDA performed in 10X Genomics platform have a better performance than regular MDA performed in tubes/wells. But, the claim could still be a bit more careful and specific in order to be rigorous, such as the 10X Genomics Single Cell CNV platform has been demonstrated a reduced noise relative to DOP-PCR, instead of directly claiming a comparison between MDA and DOP-PCR, or directly claiming a superiority in CNV detectability.

Reviewer #2 (Remarks to the Author):

I am satisfied with the authors' responses to my comments

Reviewer #3 (Remarks to the Author):

The authors have addressed my concerns. This work is important and will further the ability to analyze DNA replication in single cells.

Reviewer #1:

The authors have addressed most of my concerns and suggestions, however, there are still one major concern and one minor issue that is worth to mention for the quality of the paper.

The main concern is to follow my point 6 in the previous round, “when the authors cluster tracks into IRs by prioritizing shorter tracks over longer tracks, I wonder whether the analysis result is robust when changing such prioritizing parameters”. The authors addressed this concern by explaining in detail what “prioritize” means in their analysis pipeline, which is actually to remove longer tracks that overlapped multiple shorter tracks, indicating these are multiple firing origins, thus should not be considered. My concern is that, even after this filtering process, a certain portion of the remaining tracks could still be from multiple firing origins. Although no shorter tracks overlapping with them are observed, it does not guarantee such shorter tracks do not exist. The authors also acknowledged this possibility when addressing my point 3 in their rebuttal.

My suggestion here is that although there is no way to exactly separate tracks with single origin from those with multiple origins, at least the authors may try different filtering thresholds based on track length, in addition to the current filtering to remove tracks that are for sure from multiple origins. The goal is to make sure the most important scientific conclusions from this paper are robust and not affected by such parameter tuning in the analysis pipeline.

We thank the reviewer for this comment, articulating a caveat with which we certainly agree: we cannot rule out the possibility that additional replication initiation events occur but are not observed and/or detected in this particular dataset. We would direct the reviewer to Supplementary Figure 5, where we highlight 8 examples of IRs that appear visually to contain tracks from multiple fired origins. As can be observed in this supplementary figure, clustering together of tracks that correspond to multiple origins would disrupt the symmetry around the IR center.

We do not feel the requested analysis is informative. As described in the methods section, we do not consider track length *per se* in the identification of IRs. Rather, track clustering begins with the shortest tracks and continues until inclusion of a track causes two clusters converge – thus, the “threshold” for track inclusion is locus-specific and driven by the data, not imposed from the top down. We have added a sentence in the methods section to state this explicitly.

A minor issue is to follow my point 1 in the previous round. The PNAS paper the authors cited focused on PTA method, which is different from MDA. Although in the paper Figure 2 also compared between DOP-PCR and different versions of MDA assays, the comparison was all based on 1bp bin size. When comparing uniformity, noise, or the accuracy to detect CNV, the bin size could change the result drastically. For example, in the LIANTI paper (Science 356, 189-194, 2017), whose data the PNAS paper used a lot, Figure 1E showed CV of different methods, and suggested MDA is better than DOP-PCR in terms of uniformity/noise when bin size is <10Kb (same as PNAS paper Fig 2B), but worse than DOP-PCR when bin size is >10 Kb. That is, MDA

could be better or worse than DOP-PCR, depending on later analysis of CNVs <10Kb or >10Kb. Usually in single-cell RT studies, most CNVs (replication tracks) will be >10Kb. With that said, I do understand the authors need to claim MDA is better than DOP-PCR, since DOP-PCR has already been used to study single-cell RT before. Also, MDA performed in 10X Genomics platform have a better performance than regular MDA performed in tubes/wells. But, the claim could still be a bit more careful and specific in order to be rigorous, such as the 10X Genomics Single Cell CNV platform has been demonstrated a reduced noise relative to DOP-PCR, instead of directly claiming a comparison between MDA and DOP-PCR, or directly claiming a superiority in CNV detectability.

To alleviate the reviewer's concern, we have toned down this sentence and removed the direct comparison to DOP-PCR. It now reads:

In addition, recent studies suggest that improved MDA protocols may have reduced noise relative to previous single-cell amplification methods. (p. 4).